# CerebraGloss: Instruction-Tuning a Large Vision-Language Model for Fine-Grained Clinical EEG Interpretation

**Wei Gu[1], Tianming Luo[1], Qiran Zhang[1], Mohan Ye[1], Xiao Shen[1], Wenxin Chen[1],**
**Yunhuan Li[1], Yichen Zhang[2], Jing Hong[3], Bao-Liang Lu[1], Wei-Long Zheng[1]***
[1]Shanghai Jiao Tong University, [2]Tsinghua University,
[3]RuiJin Hospital, Shanghai Jiao Tong University School of Medicine

## Abstract

Interpreting clinical electroencephalography (EEG) is a laborious, subjective process, and existing computational models are limited to narrow classification tasks rather than holistic interpretation. A key bottleneck for applying powerful Large Vision-Language Models (LVLMs) to this domain is the scarcity of datasets pairing EEG visualizations with fine-grained, expert-level annotations. We address this by introducing CerebraGloss, an instruction-tuned LVLM for nuanced EEG interpretation. We first introduce a novel, automated data generation pipeline, featuring a bespoke YOLO-based waveform detector, to programmatically create a large-scale corpus of EEG-text instruction data. Using this data, we develop CerebraGloss, the first model of its kind capable of unified, generative analysis—performing tasks from detailed waveform description to multi-turn, context-aware dialogue. To evaluate this new capability, we construct and release CerebraGloss-Bench, a comprehensive benchmark for open-ended EEG interpretation. CerebraGloss demonstrates strong performance, surpassing leading LVLMs, including proprietary models like GPT-5, on this benchmark and achieving a new state-of-the-art on the TUSZ seizure detection task. Models, benchmark and tools are available at `https://github.com/iewug/CerebraGloss`.

## 1 Introduction

Electroencephalography (EEG) remains a fundamental diagnostic tool in neurology, yet its clinical power is unlocked only through meticulous manual review of raw waveforms by trained specialists (Kiloh et al., 2013). This process suffers from critical limitations: it is (1) **laborious**, with experts spending hours reviewing a single recording; (2) **subjective**, leading to significant inter-observer variability; and (3) **incomplete**, as pragmatic, selective annotation leaves vast amounts of signal information unanalyzed. These challenges create a major bottleneck in patient care and motivate the need for more effective analytical tools. To facilitate a broader understanding of the clinical context, we provide a primer on clinical EEG in Appendix A.

The research community's response has evolved from traditional machine learning using hand-crafted features to deep learning models and, most recently, to large-scale self-supervised foundation models (Loh et al., 2020; Shoeibi et al., 2021; Babu et al., 2025). Despite their increasing sophistication, these models share a common limitation: they are designed to perform specialized classification on isolated tasks like seizure detection or sleep staging, lacking the ability to synthesize findings or provide a holistic, interpretive analysis. Fundamentally, the field has produced classifiers, but not yet effective interpreters.

The recent success of Large Vision-Language Models (LVLMs) (Anthropic, 2024; OpenAI, 2024; Wu et al., 2024; Bai et al., 2025) offers a transformative new paradigm. By treating EEG waveforms as a specialized visual language, we can potentially adapt these powerful models to "read" and interpret neurophysiological data with human-like nuance. This approach promises a shift from

---

*Corresponding author (weilong@sjtu.edu.cn)

narrow classifiers to comprehensive interpreters. However, a critical bottleneck has prevented this leap: the absence of large-scale datasets pairing EEG visualizations with the kind of **fine-grained, expert-level interpretations** needed for effective instruction tuning.

To this end, we introduce **CerebraGloss**, an LVLM instruction-tuned for the nuanced interpretation of clinical EEG waveforms. We overcome the data bottleneck by first developing a novel, automated pipeline that programmatically generates a massive corpus of detailed annotations directly from raw EEG signals. Using this unique data engine and Gemini 2.5 Flash (Comanici et al., 2025), we create a large-scale instruction dataset and subsequently train CerebraGloss to understand and reason about EEG images. The resulting model is the first of its kind, capable of performing not only classification but also generating detailed descriptions of waveforms, artifacts, and background rhythms, and engaging in multi-turn, context-aware dialogue—mimicking the interpretive process of a clinical expert.

Our primary contributions are:

- We pioneer a new paradigm for EEG analysis that shifts from isolated classification to unified, generative dialogue, enabling a single model to perform multi-faceted interpretation of EEG segments.

- We propose a novel data generation pipeline where a suite of custom-built analysis tools—including a pioneering YOLO-based (Redmon et al., 2016) waveform detector—is used to programmatically create a large-scale EEG-text instruction dataset.

- We successfully instruction-tune a large vision-language model, demonstrating that with specialized data, a general-purpose LVLM can be adapted to interpret complex, domain-specific visualizations like clinical EEG waveforms.

- We construct and release a novel benchmark for comprehensive EEG interpretation, comprising diverse tasks designed to evaluate a model's nuanced understanding beyond single-metric classification.

## 2 RELATED WORK

**Computational Models for Clinical EEG Interpretation.** The computational analysis of EEG began with traditional machine learning classifiers (e.g., support vector machine) on hand-crafted features (Tzallas et al., 2009; Shoeb, 2009; Alickovic et al., 2018). This was followed by deep learning models, such as convolutional neural networks (CNNs) and recurrent neural networks (RNNs), which could learn representations directly from data (Supratak et al., 2017; Chen et al., 2018; Qiu et al., 2023). More recently, foundation models for EEG have emerged, pre-trained on large-scale unlabeled data using self-supervised objectives like BERT-style (Devlin et al., 2019) masked signal modeling (Zhang et al., 2023a; Jiang et al., 2024; Wang et al., 2024) or GPT-style (Radford et al., 2018) autoregressive prediction (Cui et al., 2024). While powerful, these models are predominantly evaluated as specialized classifiers for tasks spanning both clinical applications and broader BCI domains (e.g., emotion recognition and motor imagery), lacking the holistic, interpretive capability of a human expert.

**Bridging EEG and Language.** Initial efforts to connect EEG and language have primarily followed two paths. The first category aims to learn powerful EEG representations for classification by aligning signals with text. This includes methods such as ELM-MIL (Gijsen & Ritter, 2025) and EEG-CLIP (Camaret Ndir et al., 2025) that perform coarse-grained alignment between multi-hour recordings and summary-level clinical reports. By design, these approaches are not optimized for grounding textual descriptions to specific waveform events, which is central to our generative focus. The second path explores instruction tuning, where models like NeuroLM (Jiang et al., 2025) reframe classification tasks into a multiple-choice format. While innovative, this method is fundamentally non-generative, precluding free-form output or dialogue. It is also crucial to distinguish our task—interpreting the EEG signal for its clinical significance—from the separate field of brain-to-text decoding (Mishra et al., 2025), which aims to reconstruct a user's internal speech. Given the limitations of prior work, a model capable of fine-grained, generative, and conversational interpretation of clinical EEG thus remains an open challenge.

**Domain-Specific Post-Training for LVLMs.** Domain-specific post-training adapts general-purpose LVLMs to specialized fields like food (Mohbat & Zaki, 2024; Yin et al., 2025), biomedicine (Li et al., 2023; Zhang et al., 2023b; Chen et al., 2024), and remote sensing (Zhang et al., 2024). This process typically involves a two-stage training pipeline. In the first stage, the model undergoes preliminary alignment using a large corpus of domain-specific image-caption pairs to learn fundamental visual concepts and terminology. In the second stage, the model is fine-tuned on more complex visual instruction datasets to cultivate advanced reasoning and instruction-following abilities. The creation of these instruction datasets includes applying manual rules (Mohbat & Zaki, 2024), or leveraging powerful teacher models like GPT-4 (Achiam et al., 2023) to synthesize diverse conversational and question-answering data (Li et al., 2023; Chen et al., 2024).

## 3 EEG INSTRUCTION DATA GENERATION

### 3.1 AUTOMATED PIPELINE FOR STRUCTURED ANNOTATION

The foundation of CerebraGloss is a large-scale instruction dataset. As manual annotation of detailed EEG interpretations is prohibitively expensive, we designed an automated pipeline which takes **raw multi-channel EEG signals** as input and programmatically generates a set of structured clinical annotations. This pipeline serves as a "data engine", comprising three core modules for identifying significant waveform events, characterizing background activity, and detecting artifacts.

**Key Waveform Event Detection.** Central to EEG interpretation is the identification of specific, transient graphoelements. To automate this process, we developed **CerebraGloss-YOLO**, a bespoke object detection model tailored for localizing and classifying salient events within multi-channel time-series data. It is designed to recognize nine clinically critical waveform types: spikes, sharp waves, spike/sharp-and-slow-wave complexes, K-complexes, sleep spindles, high-frequency noise, positive sharp transients (blinks), positive and negative square waves (lateral eye movements). Visual examples of these waveforms are provided in Figure 5. Our team of trained annotators undertook an extensive, multi-month labeling process, meticulously curating a dataset from public corpora including DREAMS (Devuyst, 2005) and select subsets of the TUH EEG Corpus (Obeid & Picone, 2016), in addition to our private in-house collection. This effort produced a dense dataset of 46,258 expert-labeled bounding boxes across 2,849 unique 10-second EEG segments. The architecture and implementation details of CerebraGloss-YOLO are provided in Appendix B.

**Background Rhythm Characterization.** Beyond discrete events, the pipeline assesses global background characteristics. It quantifies amplitude as half of the peak-to-peak voltage and determines the dominant frequency by first identifying the canonical frequency band (i.e., delta, theta, alpha, beta, and gamma) with the highest power spectral density, and then extracting the frequency with the peak magnitude within that band.

**Artifact Identification.** To ensure robust analysis, the pipeline incorporates a module to identify common artifacts based on their statistical and morphological signatures. This module identifies physiological artifacts, such as muscle activity (EMG) via high-frequency power, eye movements (EOG) through spatial correlation patterns in frontal channels, and respiration by its rhythmic slow-wave morphology. It also flags non-physiological artifacts: electrode noise is identified by a composite criterion of extreme local amplitude combined with a loss of correlation with adjacent channels, while flat lines are marked by periods of near-zero signal variance, which may indicate either an artifact or a clinically significant low-voltage state.

### 3.2 INSTRUCTION-FOLLOWING DATA GENERATION

Leveraging the structured annotations from our pipeline, we constructed a large-scale, multi-format instruction-following dataset. We sourced a total of 1.4M 10-second EEG segments (3,889 hours), including the training sets of TUAB (Lopez et al., 2015), TUEV (Harati et al., 2015), and TUSZ (Shah et al., 2018), the entirety of TUAR (Buckwalter et al., 2021), TUEP (Veloso et al., 2017), and TUSL (von Weltin et al., 2017) (which do not provide train/test splits), along with DREAMS (Devuyst, 2005) and the first 100 subjects from HMC (Alvarez-Estevez & Rijsman, 2021;

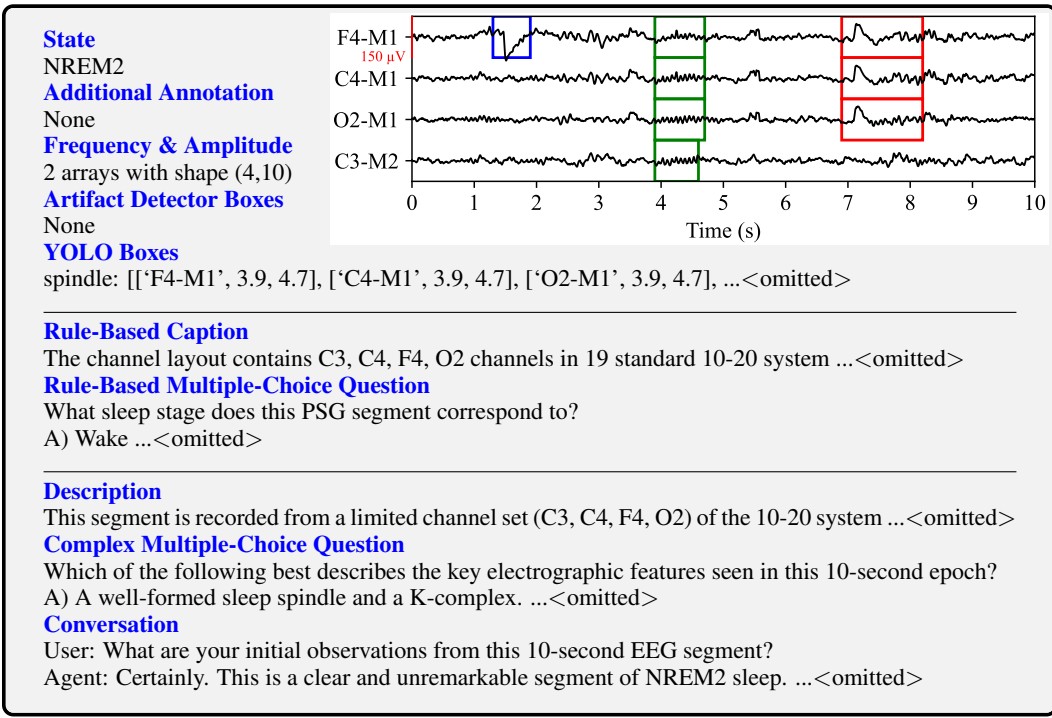

**State**
NREM2
**Additional Annotation**
None
**Frequency & Amplitude**
2 arrays with shape (4,10)
**Artifact Detector Boxes**
None
**YOLO Boxes**
spindle: [['F4-M1', 3.9, 4.7], ['C4-M1', 3.9, 4.7], ['O2-M1', 3.9, 4.7], ...<omitted>

**Rule-Based Caption**
The channel layout contains C3, C4, F4, O2 channels in 19 standard 10-20 system ...<omitted>
**Rule-Based Multiple-Choice Question**
What sleep stage does this PSG segment correspond to?
A) Wake ...<omitted>

**Description**
This segment is recorded from a limited channel set (C3, C4, F4, O2) of the 10-20 system ...<omitted>
**Complex Multiple-Choice Question**
Which of the following best describes the key electrographic features seen in this 10-second epoch?
A) A well-formed sleep spindle and a K-complex. ...<omitted>
**Conversation**
User: What are your initial observations from this 10-second EEG segment?
Agent: Certainly. This is a clear and unremarkable segment of NREM2 sleep. ...<omitted>

Figure 1: One example to illustrate the instruction-following data. "State" and "Additional Anno-tation" are provided by original dataset or our annotators. Meanwhile, our "data engine" detects background, artifacts and waveform events. Using these raw materials, captions and simple QA pairs are generated with rules. Finally, all materials except for the two background arrays and the simple QA are fed to the LLM, resulting in three types of instruction-following data. Note that the visual image is not used anywhere in the process; we only show it here as a reference.

2022). From this extensive pool, we generated instruction data through a two-pronged strategy: a systematic rule-based approach followed by augmentation with a large language model.

**Rule-Based Generation.** We first employed a programmatic approach to generate a foundational set of detailed captions and simple question-answer pairs. The template-driven captions synthesize a comprehensive description covering five key aspects: (1) montage configuration, (2) artifacts (e.g., blinks, high-frequency noise), (3) sleep-related events (e.g., K-complexes, spindles), (4) epilepti-form activity, including an assessment of dipole characteristics, and (5) background characteristics, detecting posterior dominant rhythm, paroxysmal delta rhythm and any spatial asymmetries or tem-poral variations. To mitigate the inaccuracies introduced by automated identification, strategies such as event priority masking, spatial pruning of isolated events and the inductive integration of event groups are employed. Concurrently, we generated multiple-choice and binary questions to probe for specific knowledge across key domains like artifact presence, sleep staging, and seizure detection.

**LLM-Powered Data Augmentation.** To elevate the complexity and conversational nature of our dataset, we utilized Gemini 2.5 Flash (Comanici et al., 2025)—chosen for its optimal balance of capability and cost—as a teacher model. We provided the model with the rule-based captions, the bounding boxes from CerebraGloss-YOLO and our artifact detectors, the sleep stage and additional annotation as input. Using meticulously engineered one-shot prompts, with distinct sets tailored for sleep and epileptic seizure data, we guided the model not only to generate a rich mixture of instruction types but also to constrain its output to the provided context, thereby mitigating the risk of factual inaccuracies. This process yielded a final dataset of 94K high-quality examples, balanced in a 1:1:1 ratio across three formats: (1) **Description**: a comprehensive, free-text interpretation; (2) **Complex Multiple-Choice Question**: multi-choice questions requiring deeper reasoning; and (3) **Conversation**: conversational exchanges mimicking a clinical consultation. Examples of each data type are shown in Figure 11, and the prompts are detailed in Appendix D.

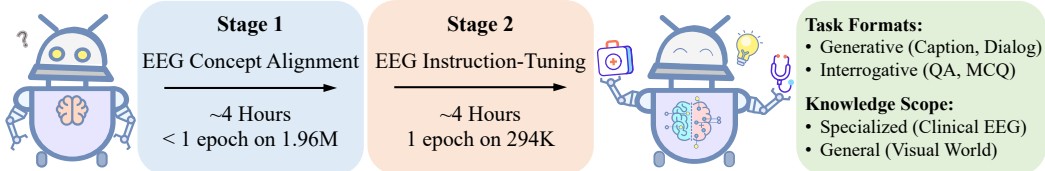

Figure 2: The two-stage training pipeline for CerebraGloss. A mix of EEG and general-domain data is used to mitigate catastrophic forgetting. Stage 1 aligns EEG visual concepts in under one epoch via early stopping, while Stage 2 fine-tunes for instruction-following. The process yields a specialized model capable of diverse generative and interrogative EEG interpretation tasks.

## 4 METHODOLOGY

To endow a general-purpose LVLM with the specialized ability to interpret clinical EEG waveforms, we perform continued post-training on the Qwen2.5-VL-3B model (Bai et al., 2025), specifically selected to leverage its robust pre-trained visual representations as a foundation for medical domain adaptation. Our primary goal is to instill fine-grained EEG understanding while preserving the model's extensive pre-trained knowledge of general visual concepts. We retain the original model architecture, which consists of a visual encoder, an LLM decoder, and a projector that bridges the two modalities. Our training curriculum for CerebraGloss follows a two-stage strategy, designed to first establish a foundational understanding of EEG concepts and then cultivate advanced instruction-following and reasoning capabilities. This process is illustrated in Figure 2, and the model structure is detailed in Appendix G.

**Stage 1: EEG Concept Feature Alignment.** The initial stage aims to align the visual features of EEG waveforms with their corresponding semantic concepts in the language model's embedding space. To achieve this while safeguarding the model's pre-existing abilities, we curate a blended dataset. This includes 1.4M EEG image-caption pairs generated programmatically by our data engine, combined with the 558K general-domain image-caption pairs from the LLaVA Visual Instruct Pretrain LCS-558K (Liu et al., 2024a). During this stage, we freeze the parameters of both the visual encoder and the LLM decoder, exclusively performing full-parameter fine-tuning on the projector. This targeted approach efficiently teaches the model the new visual vocabulary of EEG without risking catastrophic forgetting. We employ an early stopping strategy, concluding the training phase before the convergence point of the training loss curve, thereby preventing overfitting and optimizing training time.

**Stage 2: EEG Instruction-Tuning.** The second stage focuses on developing the model's ability to follow complex instructions, generate nuanced interpretations, and engage in conversational dialogue. For this, we construct a diverse instruction-following dataset comprising both domain-specific and general-purpose examples. The EEG-specific data includes 100K rule-based multiple-choice questions (containing 39K TUSZ seizure questions and 40K HMC sleep staging questions), 94K instruction samples (covering multi-turn conversations, detailed descriptions, and complex reasoning questions) generated by Gemini 2.5 Flash, and an additional 50K rule-based captions from the HMC and TUSZ datasets. To maintain general instruction-following capabilities, we supplement this with 50K general-domain samples, consisting of 30K from the CoSyn-400K (Yang et al., 2025) and 20K from the LLaVA-Instruct-150K (Liu et al., 2023). In this stage, we freeze the visual encoder and perform full-parameter fine-tuning on both the LLM decoder and the projector. The model is trained for a single epoch on this combined dataset.

**Implementation Details.** We conducted all training experiments on a cluster of 8 NVIDIA A800 (80GB) GPUs. We employed the AdamW optimizer with an effective batch size of 256, achieved through gradient accumulation. The learning rate was managed by a cosine scheduler with a peak value of $1 \times 10^{-5}$ and a warmup ratio of 0.1. The entire two-stage training process is highly efficient; with the early stopping strategy in the first stage, each stage was completed in approximately 4 hours. More information about instructions and data format of training are detailed in Appendix F.

# 5 CEREBRAGLOSS-BENCH: A BENCHMARK FOR NUANCED EEG INTERPRETATION

Existing clinical EEG benchmarks are limited to closed-set classification tasks, such as seizure detection in TUSZ (Shah et al., 2018) or sleep staging in HMC (Alvarez-Estevez & Rijsman, 2021; 2022). While valuable, this paradigm is insufficient for evaluating nuanced interpretation. Specifically, this single-label approach creates a label-granularity mismatch by incorrectly propagating file-level labels to every segment, oversimplifies complex signals that may contain multiple co-occurring events, and ignores crucial context-dependency where a waveform's meaning changes with patient state. A detailed discussion of these issues is provided in Appendix C.

To address these limitations and to rigorously evaluate a model's ability to "read" EEG, we introduce and will publicly release **CerebraGloss-Bench**. To our knowledge, it is the first benchmark designed for *open-ended* clinical EEG interpretation and *multi-class waveform object detection*. CerebraGloss-Bench comprises 90 challenging 10-second segments of full 19-channel 10-20 system EEG. Each segment is paired with a four-part evaluation suite: a free-text **description**, a complex **multiple-choice question (MCQ)**, a conversational **question-answer pair (QA)**, and dense, channel-level **bounding box annotations** for nine critical waveform types (detailed in Section 3.1). The textual data was initially generated using a programmatic prompting strategy and subsequently reviewed, edited, and validated by clinical experts to ensure high quality and accuracy. All data was sourced

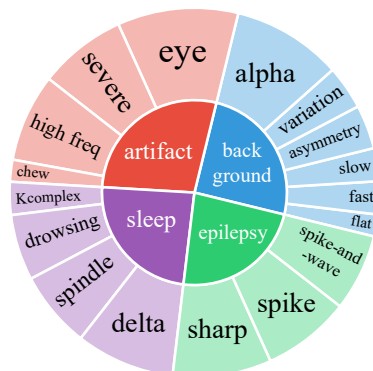

Figure 3: Distribution of test topics in CerebraGloss-Bench

from a private in-house collection, with subjects entirely disjoint from those used in our training data to prevent data leakage and ensure a fair evaluation. The benchmark offers comprehensive coverage of clinically relevant phenomena, spanning four major categories and seventeen sub-categories: background rhythms (alpha rhythm, temporal variation, spatial asymmetry, slowing, fast activity, low voltage), artifacts (eye-related, severe artifact, high-frequency noise, chewing), sleep patterns (K-complexes, drowsing slow activity, sleep spindles, delta activity in deep sleep), and epileptiform patterns (sharp waves, spikes, and spike/sharp-and-slow-wave complexes). The distribution of these assessment areas is shown in Figure 3, and examples are presented in Appendix N.

# 6 EXPERIMENTS

## 6.1 QUALITATIVE ANALYSIS

Table 1 shows CerebraGloss following diverse instructions to generate clinically relevant interpretations. It correctly identifies the primary feature (a posterior dominant alpha rhythm), localizes transient artifacts, and synthesizes this information coherently. Despite these promising capabilities, the model is not without limitations. We must admit that CerebraGloss still has room for improvement, particularly in mitigating occasional false positives where it hallucinates non-existent waveforms—a challenge likely stemming from noise in our automated data generation pipeline. A detailed discussion of noise propagation is presented in Appendix I. Additional results and expert-annotated ground truth can be found in Appendix N.

## 6.2 MAIN RESULTS

**CerebraGloss-Bench.** We evaluated CerebraGloss against state-of-the-art multimodal large language models including specialized biomedical ones on CerebraGloss-Bench. We report accuracy for multiple-choice questions. For open-ended tasks, we use ROUGE-1 (Lin, 2004) to measure factual recall in descriptions and, following prior work (Liu et al., 2023; Li et al., 2023), use GPT-5 (OpenAI, 2025) as an impartial judge for conversational QA. GPT-5 scores the model's response on a 1-to-10 scale across helpfulness, relevance, accuracy, and detail by comparing it against the expert-authored ground truth. As shown in Table 3, CerebraGloss achieves state-of-the-art perfor-

Table 1: CerebraGloss accurately follows the user's instructions. Note that the EEG is shown in two images to conserve space in the layout. The model is fed the full, single image.

**EEG input example, posterior dominant alpha rhythm**

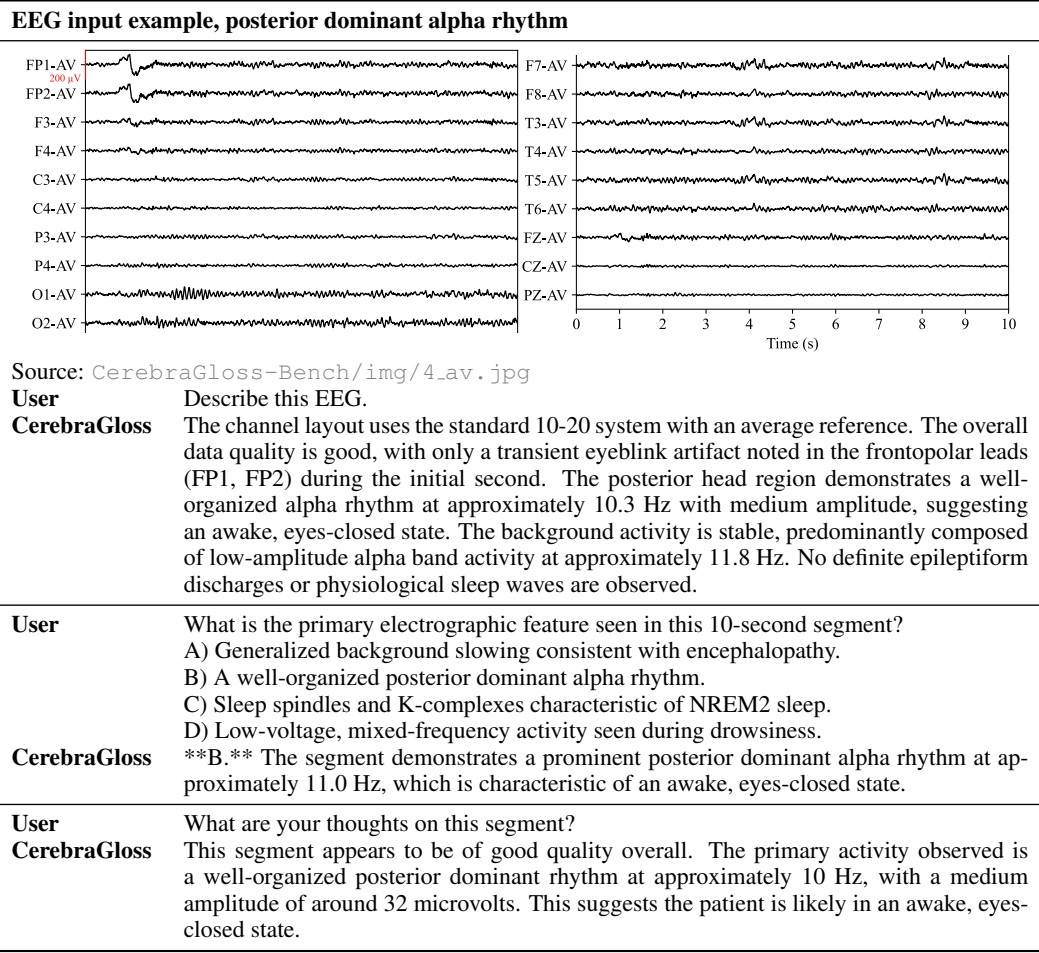

Source: `CerebraGloss-Bench/img/4_av.jpg`

| | |
|---|---|
| **User** | Describe this EEG. |
| **CerebraGloss** | The channel layout uses the standard 10-20 system with an average reference. The overall data quality is good, with only a transient eyeblink artifact noted in the frontopolar leads (FP1, FP2) during the initial second. The posterior head region demonstrates a well-organized alpha rhythm at approximately 10.3 Hz with medium amplitude, suggesting an awake, eyes-closed state. The background activity is stable, predominantly composed of low-amplitude alpha band activity at approximately 11.8 Hz. No definite epileptiform discharges or physiological sleep waves are observed. |
| **User** | What is the primary electrographic feature seen in this 10-second segment?
A) Generalized background slowing consistent with encephalopathy.
B) A well-organized posterior dominant alpha rhythm.
C) Sleep spindles and K-complexes characteristic of NREM2 sleep.
D) Low-voltage, mixed-frequency activity seen during drowsiness. |
| **CerebraGloss** | **B.** The segment demonstrates a prominent posterior dominant alpha rhythm at approximately 11.0 Hz, which is characteristic of an awake, eyes-closed state. |
| **User** | What are your thoughts on this segment? |
| **CerebraGloss** | This segment appears to be of good quality overall. The primary activity observed is a well-organized posterior dominant rhythm at approximately 10 Hz, with a medium amplitude of around 32 microvolts. This suggests the patient is likely in an awake, eyes-closed state. |

mance, surpassing even powerful proprietary models like GPT-5. Notably, models such as LLaVA-Med (Li et al., 2023) and BioMedGPT (Luo et al., 2024), despite their biomedical focus, struggle significantly on this benchmark. Their training corpora lack paired EEG-text data, rendering them unable to interpret these specialized visualizations. This underscores the critical importance of in-domain instruction tuning and demonstrates the efficacy of our approach.

**Waveform Detection.** The benchmark's waveform detection task evaluates localization and classification. We assessed CerebraGloss-YOLO using mean Average Precision (mAP) at an Intersection-over-Union (IoU) of 0.5. As this is the first benchmark for multi-class, channel-wise waveform detection, direct comparisons are unavailable. CerebraGloss-YOLO achieves a promising mAP (Table 2), establishing a strong baseline for this new task.

Table 2: Waveform detection

| Model | mAP@0.5 |
|---|---|
| CerebraGloss-YOLO | 40.95% |

**Standard Clinical Tasks.** To validate CerebraGloss on established benchmarks, we assessed its performance on two standard clinical classification tasks: seizure detection and sleep staging. For seizure detection, we used TUSZ (Shah et al., 2018), which provides labels for seizure and non-seizure periods. For sleep staging, we utilized the HMC (Alvarez-Estevez & Rijsman, 2021; 2022), which originally contains five stages (Wake, NREM1, NREM2, NREM3, and REM) labeled in 30-second epochs. To align with our model's architecture, we standardized the input by segmenting all recordings into non-overlapping 10-second epochs. For HMC, each 30-second label was assigned

Table 3: Instruction-following capability comparison on CerebraGloss-Bench. Multiple-choice questions (MCQ), descriptions, and question-answering (QA) are evaluated using accuracy (%), ROUGE-1 score (%), and GPT-5 score (1-10), respectively. CerebraGloss even outperforms GPT-5. LLaVA-Med and BioMedGPT cannot follow instructions for MCQs.

| | MCQ | Description | QA |
|---|---|---|---|
| LLaVA-Med (Li et al., 2023) | / | 8.87 | 2.83 |
| BioMedGPT (Luo et al., 2024) | / | 11.82 | 1.29 |
| Qwen2.5-VL-32B (Bai et al., 2025) | 37.78 | 36.90 | 3.57 |
| Gemini 2.5 Pro (Comanici et al., 2025) | 52.22 | 37.95 | 3.86 |
| GPT-5 (OpenAI, 2025) | 70.00 | 37.07 | 4.58 |
| CerebraGloss-3B | **80.00** | **44.19** | **4.76** |

Table 4: Balanced accuracy (%) on TUSZ and HMC. CerebraGloss significantly outperforms its base model Qwen2.5-VL-3B and achieves a new SOTA result on TUSZ. ELM-MIL cannot be tested on HMC due to its montage setting.

| Model | Type | Multi-task | TUSZ | HMC |
|---|---|---|---|---|
| EEGNet (Lawhern et al., 2018) | DL | ✗ | 65.53 | 58.51 |
| CNN-Transformer (Peh et al., 2022) | DL | ✗ | 75.53 | 68.35 |
| ELM-MIL (Gijsen & Ritter, 2025) | DL+ML | ✗ | 78.27 | / |
| LaBraM (Jiang et al., 2024) | LEM | ✗ | 77.48 | 68.92 |
| Gram (Li et al., 2025) | LEM | ✗ | 78.29 | **69.97** |
| LLaVA-Med (Li et al., 2023) | LVLM | ✓ | 50.00 | 25.00 |
| BioMedGPT (Luo et al., 2024) | LVLM | ✓ | 50.00 | 25.00 |
| Qwen2.5-VL-3B (Bai et al., 2025) | LVLM | ✓ | 55.02 | 25.00 |
| CerebraGloss-3B | LVLM | ✓ | **79.21** | 62.02 |

to the three corresponding 10-second segments. Furthermore, since the definitive criteria for REM sleep rely on electromyography (EMG) and electrooculography (EOG) signals—modalities not used by our model—we excluded the REM stage, formulating the task as a four-class classification. To ensure a rigorous evaluation with no subject overlap from the training set, we used the official evaluation split of TUSZ (46,091 samples) and the final 26 subjects from HMC (60,678 samples).

We benchmarked CerebraGloss against classic deep learning (DL) architectures, state-of-the-art large EEG models (LEMs), LVLMs and the most recent work ELM-MIL that combines EEG and clinical report. Given the significant class imbalance in both datasets, we report balanced accuracy as the primary evaluation metric. The results are summarized in Table 4. CerebraGloss achieves a new state-of-the-art on the TUSZ seizure detection task, outperforming all specialized models. In contrast, the LVLMs demonstrate negligible performance, often defaulting to repetitive answers and thus achieving only chance-level accuracy. On the HMC sleep staging task, CerebraGloss's performance is competitive yet falls slightly below the top-performing LEM. We posit this discrepancy is less a limitation of our model's interpretive ability and more a reflection of the task's specific demands. Clinical sleep staging often requires temporal context spanning several minutes to resolve ambiguities. Specialized models, designed to excel at this singular task, may be highly tuned to subtle, short-segment patterns that help distinguish between similar sleep stages. CerebraGloss, in contrast, is trained for a broader, more descriptive interpretation, which may naturally de-emphasize optimization for a single, context-poor classification task. A more detailed analysis for HMC is presented in Appendix H.

**General Capabilities.** In addition to specialized clinical performance, we evaluated whether our fine-tuning process compromises the model's general vision-language abilities. We assessed CerebraGloss-3B on the comprehensive MMBench (Liu et al., 2024b) benchmark and found that it retains its core capabilities with only a marginal performance decrease compared to the original Qwen2.5-VL-3B, demonstrating that our approach successfully avoids significant catastrophic forgetting. The detailed results and analysis are provided in Appendix K.

Table 5: Ablation studies on training configuration and model scale. It evaluates the impact of training duration, Stage 2 data composition (1 epoch but without captions or without LLM-augmented data), and model size.

| Model Variants | | | Clinical Tasks | | CerebraGloss-Bench | | |
|---|---|---|---|---|---|---|---|
| Params | Stage 1 | Stage 2 | TUSZ | HMC | MCQ | Description | QA |
| 3B | 0 | 1 | 79.68 | **62.24** | 78.89 | 41.11 | 4.57 |
| 3B | 0.05 | 1 | 79.21 | 62.02 | **80.00** | **44.19** | **4.76** |
| 3B | 0.10 | 1 | **79.83** | 61.46 | 76.67 | 41.03 | 4.40 |
| 3B | 0.20 | 1 | 79.23 | 61.16 | 74.44 | 41.69 | 4.30 |
| 3B | 0.05 | 0 | 54.36 | 24.09 | 37.78 | 22.08 | 2.67 |
| 3B | 0.05 | 0.04 | 53.32 | 29.95 | 51.11 | 42.66 | 3.13 |
| 3B | 0.05 | 0.25 | 80.03 | 56.26 | 76.66 | 40.10 | 4.22 |
| 3B | 0.05 | 0.50 | 78.66 | 60.74 | 77.78 | 43.84 | 4.40 |
| 3B | 0.05 | w/o cap | 78.73 | 61.80 | 78.89 | **51.09** | 4.58 |
| 3B | 0.05 | w/o aug | 78.39 | 61.29 | 47.78 | 9.02 | 2.34 |
| 7B | 0.06 | 1 | **80.21** | **63.34** | **81.11** | **44.23** | 4.64 |

## 6.3 ABLATION STUDIES

We conducted a series of ablation studies to validate our key design choices, including the training data configuration and model scale. All results are presented in Table 5.

**Impact of Stage 1 Feature Alignment.** We first investigated the impact of the Stage 1 feature alignment by comparing four checkpoints: skipping Stage 1 entirely (0 epochs), an early underfitting point (0.05 epochs, see Appendix J), the training loss elbow point (0.1 epochs), and a near-overfitting point (0.2 epochs). While performance on the TUSZ and HMC classification tasks remains comparable across all settings, the model trained to the underfitting point (0.05 epochs) achieves the best results on all three generative CerebraGloss-Bench tasks. We hypothesize that this early checkpoint is optimal because it allows the model to acquire the essential visual vocabulary of EEG without overwriting its powerful, pre-existing reasoning capabilities. Further training on our programmatically generated, template-heavy captions may introduce a "descriptive bias", which hinders performance on more complex, open-ended reasoning tasks.

**Impact of Stage 2 Data Composition.** With the optimal Stage 1 configuration, we observed that model performance scaled positively with the amount of Stage 2 instruction data before converging, confirming the value of our dataset. We further explored the role of data components by removing the 50K rule-based captions from the Stage 2 mixture. This reveals an interesting trade-off: performance on the benchmark description task improved, while scores on other tasks decreased. We hypothesize that the simpler, rule-based captions, though of lower quality than the LLM-generated data, act as a form of regularization. They anchor the model's understanding to a broader, more fundamental feature space, preventing it from over-specializing on the stylistic nuances of the LLM-generated text. Removing them allows the model to better mimic the high-quality description style required by the benchmark, but at the cost of the general reasoning capabilities inherited from the base model. Additionally, we ablated the 94K instruction-following data generated by Gemini and found that the model loses its open-ended generative ability, defaulting to the MCQ format or producing gibberish filled with options. This occurs because the remaining Stage 2 data are almost entirely MCQ-based. However, performance on HMC and TUSZ does not decline, as their learning relies on rule-generated MCQs.

**Impact of Model Scale.** Finally, we investigated the scalability of our approach by applying our optimal training configuration to a larger, 7B parameter version of the model. The 7B model shows a general trend of improvement across the evaluation metrics, enhancing performance on both standard clinical tasks and most aspects of CerebraGloss-Bench. While we observed a minor decrease in the QA score, the overall positive scaling confirms that our data generation and training pipeline is effective and suggests that performance can be further enhanced by leveraging larger base models.

## 7 CONCLUSION

In this work, we introduced CerebraGloss, a pioneering LVLM that reframes automated EEG analysis from narrow classification to comprehensive, generative interpretation. We overcame the critical data bottleneck by developing a novel programmatic pipeline, featuring the CerebraGloss-YOLO detector, to create a large-scale instruction-following dataset. Through a specialized two-stage training curriculum, CerebraGloss establishes a new paradigm for unified EEG analysis via generative dialogue. To evaluate this capability, we also built and released CerebraGloss-Bench, the first benchmark for open-ended EEG interpretation and multi-class waveform object detection. Our experiments show that CerebraGloss not only sets a new state-of-the-art on the TUSZ seizure detection task but also surpasses powerful proprietary models on our novel interpretive benchmark.

While CerebraGloss establishes a new performance baseline, this work also charts a course for future research. Our image-based approach intentionally mirrors current clinical practice; however, a paradigm shift towards direct signal-to-text modeling represents a more ambitious and potentially powerful frontier. This, along with avenues for enhancing our data pipeline, extending temporal reasoning, and ensuring clinical readiness through rigorous validation, constitutes the next wave of challenges. We provide a detailed discussion of these future directions in Appendix L. By open-sourcing our model, benchmark, and tools, we aim to equip the research community with the foundational tools to pursue these exciting frontiers and accelerate the development of truly assistive neuro-intelligent systems.

## ACKNOWLEDGMENTS

This work was supported in part by grants from Brain Science and Brain-like Intelligence Technology-National Science and Technology Major Project (2025ZD0218900), National Key Research and Development Program of China (2024YFC3606800), National Natural Science Foundation of China (62376158), STI 2030-Major Projects+2022ZD0208500, Medical-Engineering Interdisciplinary Research Foundation of Shanghai Jiao Tong University "Jiao Tong Star" Program (YG2023ZD25, YG2024ZD25 and YG2024QNA03), Shanghai Jiao Tong University 2030 Initiative, the Lingang Laboratory (Grant No. LGL-1987), GuangCi Professorship Program of RuiJin Hospital Shanghai Jiao Tong University School of Medicine, and Shanghai Jiao Tong University SCS-Shanghai Emotionhelper Technology Co., Ltd Joint Laboratory of Affective Brain-Computer Interfaces.

## ETHICS STATEMENT

**Intended Use and Limitations.** We state unequivocally that CerebraGloss and our data engine including CerebraGloss-YOLO are research prototypes, **intended strictly for non-commercial, academic purposes**. As such, it is **not intended for clinical diagnosis, patient care, or any real-world medical decision-making**. The model is designed to assist researchers in analyzing EEG data and to spur further investigation into general-purpose neuro-intelligent systems. As with any generative model, CerebraGloss is susceptible to generating factually incorrect information or "hallucinations". This risk is compounded by the fact that it was trained on data from a fully automated pipeline, which, despite its effectiveness, can introduce labeling noise or errors. Therefore, its outputs must be critically reviewed by qualified clinical experts and should never be used as a substitute for professional medical judgment.

**Data Privacy and Governance.** Our training data includes both public, de-identified datasets (e.g., TUH, DREAMS, HMC) and a private, in-house data collection. All data from the private collection were fully anonymized and collected under protocols approved by an institutional review board, with informed consent obtained from all participants. Our newly created benchmark, CerebraGloss-Bench, was also sourced from this ethically approved and anonymized private collection and contains no personally identifiable information.

**Broader Impact.** Our immediate goal is to accelerate research by providing powerful, open-source tools for EEG analysis and **encourage the computational community to ground their**

**innovations in the inherent clinical and neuroscientific value of EEG signals**. By releasing our model, benchmark, and the data generation engine, we hope to foster a collaborative and transparent research environment. While we must reiterate that the current version of CerebraGloss is strictly a research prototype, the long-term vision that motivates this work is the development of reliable AI assistants for neurology. We envision a future where such systems could support clinicians by automating routine analysis, highlighting potential areas of concern for expert review, and reducing inter-observer variability in EEG interpretation.

## REPRODUCIBILITY STATEMENT

The code for data engine, the model weights for CerebraGloss, and the complete CerebraGloss-Bench are released at `https://github.com/iewug/CerebraGloss`. Details of our two-stage training methodology, including hyperparameters, are described in Section 4. The architecture of CerebraGloss-YOLO is detailed in Appendix B. We have chosen to release our data engine rather than the specific 1.4M generated data instances, mainly because our data was derived from publicly available sources (listed in Section 3.2). Releasing the engine empowers other researchers to apply our pipeline to their own private or specific EEG collections, granting them full control over the data generation process and its outputs.

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

# A  CLINICAL EEG PRIMER

## A.1  EEG ACQUISITION AND SIGNAL REPRESENTATION

Electroencephalography (EEG) is a non-invasive neurophysiological technique that measures the electrical activity of the brain via electrodes placed on the scalp. The resulting data is a multi-channel time-series signal, where each channel represents the voltage difference between two points over time. This high temporal resolution makes EEG an invaluable tool for capturing transient neural events.

**Recording Setups**: The most common standard for clinical and research applications is the International 10-20 System, which provides a standardized method for placing 19 recording electrodes and 2 reference electrodes across the scalp, as illustrated in Figure 4. Another common setup is Polysomnography (PSG), or a sleep study, which typically uses a smaller subset of EEG channels (e.g., central and occipital) alongside other physiological sensors to monitor sleep.

**Key Signal Processing Concepts**: Raw EEG signals are typically pre-processed before analysis. This involves filtering to isolate the relevant frequency spectrum (e.g., with a 1.6-70 Hz band-pass filter) and eliminate specific environmental noise, such as 50 Hz or 60 Hz power-line interference using a notch filter. The signal is then often downsampled to a lower sampling rate (e.g., 200 Hz) to reduce computational load. Two other critical concepts are montage and re-referencing. A montage is the specific combination of channels displayed for visual review. Re-referencing is the computational process of subtracting the signal from one or more reference electrodes from all other electrodes. This is crucial for mitigating widespread noise and highlighting focal brain activity.

**Our Approach**: All raw EEG signals are first band-pass filtered between 1.6-70 Hz and notch-filtered at 50/60 Hz. The signals are then downsampled to 200 Hz. For full 19-channel recordings from the 10-20 system, we employ an average reference. For the few-channel EEG data from PSG, we use a standard ear or mastoid reference (e.g., C3-A2). This scheme is critical for our model as it provides a consistent polarity representation for key events across the scalp. For instance, widespread artifacts like eye blinks consistently appear as positive deflections in frontal channels, while epileptiform discharges are typically represented as negative-going waves. This uniformity simplifies the feature space, allowing the model to more easily learn the spatial signatures of different events, a task complicated by other referencing schemes (e.g., bipolar) where polarity can reverse between adjacent channels. Additionally, a ten-second non-overlapping division scheme is adopted for all the samples we use in the study.

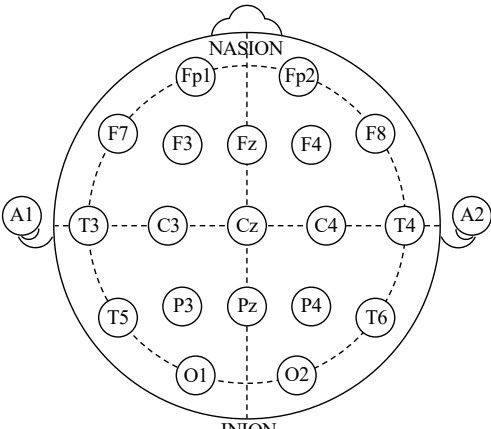

Figure 4: The International 10-20 System. The diagram illustrates the standardized placement of the 19 recording electrodes and two reference electrodes (A1 and A2) that constitute the canonical 10-20 system. The electrode names correspond to their scalp location: Fp (Frontopolar), F (Frontal), C (Central), T (Temporal), P (Parietal), and O (Occipital). We emphasize this formal definition, as the term "10-20 system" is sometimes inaccurately used in the literature to describe various electrode subsets derived from the higher-density 10-10 system.

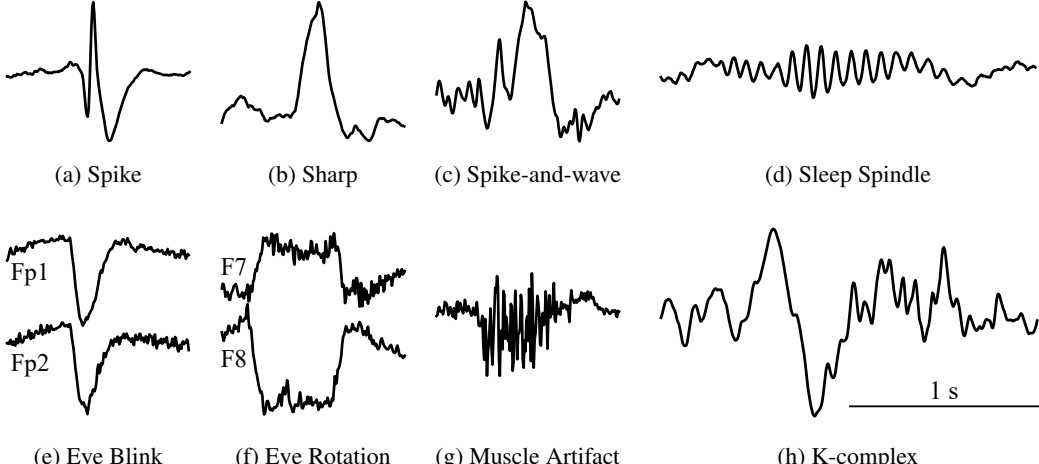

(a) Spike      (b) Sharp      (c) Spike-and-wave      (d) Sleep Spindle

(e) Eye Blink      (f) Eye Rotation      (g) Muscle Artifact      (h) K-complex

Figure 5: A visual vocabulary of key EEG waveforms. This figure displays snippets of common patterns encountered in clinical EEG interpretation. The temporal scales of the eight subplots remain uniform, while the amplitude scales have been adjusted for enhanced display clarity.

## A.2 THE "VOCABULARY" OF EEG: KEY WAVEFORMS AND PATTERNS

Clinical experts typically evaluate an EEG by analyzing its background rhythms, identifying key graphoelements (significant waveforms), and distinguishing them from artifacts. To illustrate the key patterns our model is trained to interpret, we present a collection of representative waveform snippets in Figure 5.

**Background Rhythms**: The ongoing background activity of the EEG is characterized by several frequency bands, each associated with different brain states:

- **Delta** ($\delta$, 0.3-3.5 Hz): Predominant during deep sleep in adults or indicative of brain injury.
- **Theta** ($\theta$, 4-7.5 Hz): Associated with drowsiness, light sleep, and some cognitive processes.
- **Alpha** ($\alpha$, 8-13 Hz): The hallmark of a relaxed, wakeful state with eyes closed, typically strongest over the posterior regions.
- **Beta** ($\beta$, 14-30 Hz): Common in an alert, active, or anxious state, and can be induced by certain medications.
- **Gamma** ($\gamma$, >30 Hz). With little interest.

**Clinically Significant Graphoelements**: These are distinct, transient waveforms that hold significant diagnostic value.

- **Epileptiform Discharges**: These are the primary markers for a predisposition to seizures. They are transient events that stand out from the background activity. Key examples include spikes, which are very brief, high-amplitude potentials (Figure 5a); sharp waves, which have a similar morphology but a slightly longer duration (Figure 5b); and spike-and-wave complexes or sharp-and-wave complexes, where a spike or a sharp is immediately followed by a slow wave (Figure 5c).
- **Sleep Patterns**: Specific waveforms are the hallmarks of different sleep stages. The defining features of Non-Rapid Eye Movement stage 2 (NREM2) sleep include sleep spindles, which are characteristic bursts of 11-14 Hz activity (Figure 5d), and K-complexes, which are large, biphasic slow waves followed by sleep spindles (Figure 5h).

**Common Artifacts**: A major challenge in EEG interpretation is distinguishing true neural signals from artifacts, which are non-cerebral electrical potentials. Our model must learn to differentiate true signals from common contaminants such as eye blinks, which manifest as high-amplitude, synchronous vertical deflections in frontal channels (Figure 5e), and lateral eye movements, which pro-

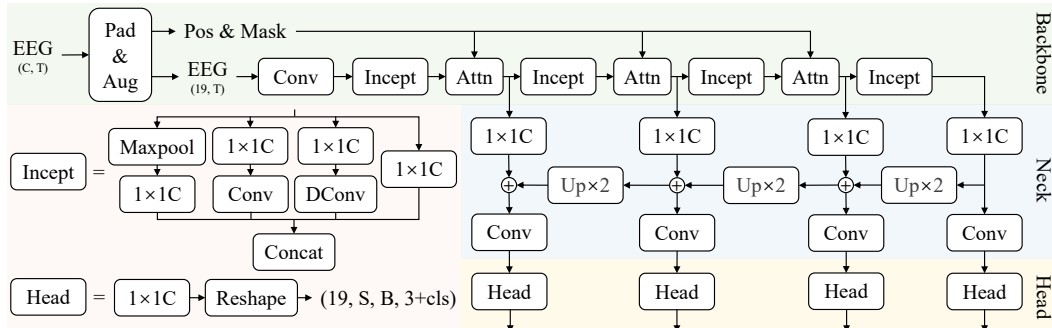

Figure 6: The CerebraGloss-YOLO structure. CerebraGloss-YOLO follows a Backbone-Neck-Head paradigm. After being padded and augmented, EEG data will go into the backbone, which is composed of inception modules (Incept), 1D convolutional layer (Conv), 1D dilated convolutional layer (DConv), pointwise convolutional layer (1×1C) and self-attention layer (Attn) to catch temporal and spatial information. Later, the neck will employ upsampling (Up×2) to fuse information from deeper layers to earlier layers. Finally, prediction heads will transform the feature map into a final prediction tensor, which is in the shape of (19 channels, S grid cells, B anchor boxes, 3+class number), where 3 is the confidence score, displacement and scaling parameters of anchor boxes. Note that batch normalization and ReLU activation are omitted for simplicity.

duce opposing slow waves in channels on opposite sides of the head (Figure 5f). Muscle activity is another frequent artifact, contaminating the signal with high-frequency, irregular noise (Figure 5g).

# B    CEREBRAGLOSS-YOLO: A CHANNEL-WISE DETECTOR FOR RAW EEG WAVEFORM DETECTION

## B.1    MODEL ARCHITECTURE

Detecting transient waveforms in raw multi-channel EEG is fundamentally a one-dimensional object detection task. However, this approach has been largely unexplored, primarily due to the scarcity of large-scale, densely annotated datasets required for training such models. The design of such a detector must address several unique challenges of EEG data: (1) events are localized to specific channels, requiring per-channel predictions; (2) channels possess a spatial relationship defined by the electrode montage, which contains clinically relevant information; and (3) events occur across a wide range of time scales, from brief spikes (70 ms) to persistent artifacts like high-frequency noise that can span the entire window. To address these issues, we developed CerebraGloss-YOLO (Figure 6), a bespoke detector inspired by YOLOv3(Redmon & Farhadi, 2018). Examples are shown in Figure 7.

**Backbone Network.**   The backbone is responsible for extracting a hierarchy of features from the input signal. To capture waveform features at multiple time scales, we employ an inception module (Szegedy et al., 2015), which uses parallel branches with different 1D convolutional kernel sizes—including dilated convolutions—to learn representations of both short- and long-duration events simultaneously. To move beyond treating channels as independent streams and explicitly model their spatial topology, we introduce a self-attention module (Vaswani et al., 2017), which treats the feature vector of each channel as a token in a sequence. By adding a learnable position embedding, we inject prior knowledge of each electrode's spatial location into the model. The self-attention mechanism then allows the model to dynamically weight and aggregate information from other channels, while an attention mask ensures it can gracefully handle missing or padded channels.

**Feature Pyramid Neck.**   Given that EEG graphoelements exhibit significant duration variability, we employ a Feature Pyramid Network (Lin et al., 2017) to create robust, multi-scale feature representations. The Neck takes feature maps from multiple stages of the backbone and fuses them via a top-down pathway with lateral connections. This process combines high-level semantic information from deeper layers with high-resolution temporal information from earlier layers. The output is a

feature pyramid where each level has a different temporal resolution, making the model adept at detecting events of varying lengths.

**Prediction Heads.** A simple prediction head, composed of a single pointwise convolution layer, can be attached to any level of the feature pyramid. This head transforms the feature map from the neck into a final prediction tensor. This tensor encodes, for each channel, temporal grid cell, and predefined anchor box, the necessary parameters for detection: bounding box coordinate offsets, an objectness confidence score, and classification logits for the nine target waveform classes.

### B.2 TRAINING AND IMPLEMENTATION DETAILS

**Data Preprocessing and Standardization.** All EEG signals used for training were segmented into 10-second clips, sampled at 200 Hz, resulting in an input tensor dimension of $(C, 2000)$, where C is the number of channels. To handle variability in recording montages, we standardized all samples to the 19-channel 10-20 international system. For recordings with fewer channels, missing channels were zero-padded; for those with more, only the standard 19 were used. Finally, each channel was independently normalized using a z-score transformation.

**Data Augmentation.** To improve model generalization and robustness to signal variations, we applied a series of augmentations during training. Standard time-series augmentations included the addition of Gaussian noise, random amplitude scaling, and random temporal circular shifts. We also employed two EEG-specific augmentations: (1) random channel dropout, where a subset of channels is zeroed out to simulate poor electrode contact, and (2) random channel permutation, where the physical order of channels in the input tensor is shuffled. This latter technique is a strong regularizer that forces the model to rely on its learned channel positional embeddings to understand spatial relationships, rather than memorizing a fixed input order.

**Anchor Design and Loss Function.** Our model employs a set of predefined 1D anchor boxes to detect events of varying durations. To match specific events to appropriate feature resolutions, we placed two shorter anchors (0.45 s and 1.5 s) on the highest-resolution feature map and one longer anchor (9.5 s) on the lowest-resolution feature map. While our architecture supports predictions at all pyramid levels, we empirically found this sparse configuration offered the best trade-off between performance and computational cost, as adding prediction heads to intermediate levels did not yield significant gains. During training, each ground-truth box is assigned to the anchor with the highest 1D Intersection-over-Union (IoU). The model is optimized using a composite loss function, standard in YOLO-based models. It consists of a mean squared error (MSE) loss for bounding box coordinate regression and binary cross-entropy (BCE) losses for the objectness score and class predictions. To balance these components, we apply distinct weights: the coordinate loss ($\lambda_{coord}$) is heavily up-weighted to prioritize accurate localization; the objectness loss for anchors containing a ground-truth object ($\lambda_{obj}$) is also boosted; and the objectness loss for background anchors ($\lambda_{noobj}$) is down-weighted to prevent the vast number of negative examples from overwhelming the training signal.

**Hyperparameters.** CerebraGloss-YOLO was trained for 80 epochs using the Adam optimizer with a learning rate of 1e-4 and a batch size of 32. The specific loss weights were set to $\lambda_{coord} = 10$, $\lambda_{obj} = 5$, and $\lambda_{noobj} = 0.5$.

## C  A CRITICAL DISCUSSION ON EEG DATASETS FOR SEGMENT-LEVEL CLASSIFICATION

Many widely used public EEG datasets, while valuable for developing models for specific applications, present inherent limitations when adapted for general-purpose, segment-level classification on short epochs (e.g., 10 seconds). This appendix elucidates these limitations, which primarily revolve around label-granularity mismatch, the oversimplification of co-occurring events, and strong dependency on external context.

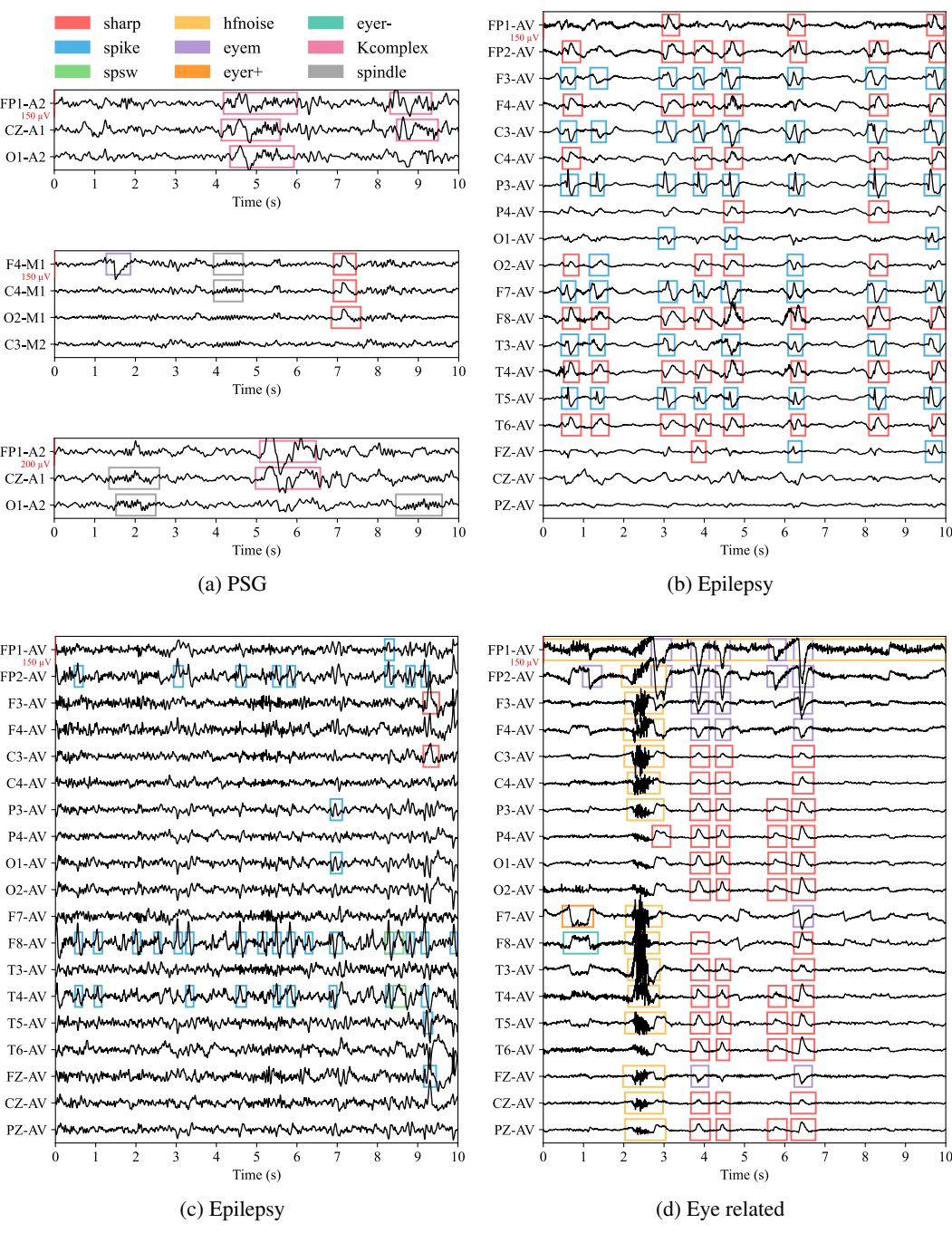

Figure 7: CerebraGloss-YOLO can perform channel-wise recognition of nine types of waveforms, including sharps, spikes, sharp/spike-and-wave complexes (spsw), high frequency noise (hfnoise), eye blinks (eyem), lateral eye movement (eyer+, eyer-), K-complexes and sleep spindles.

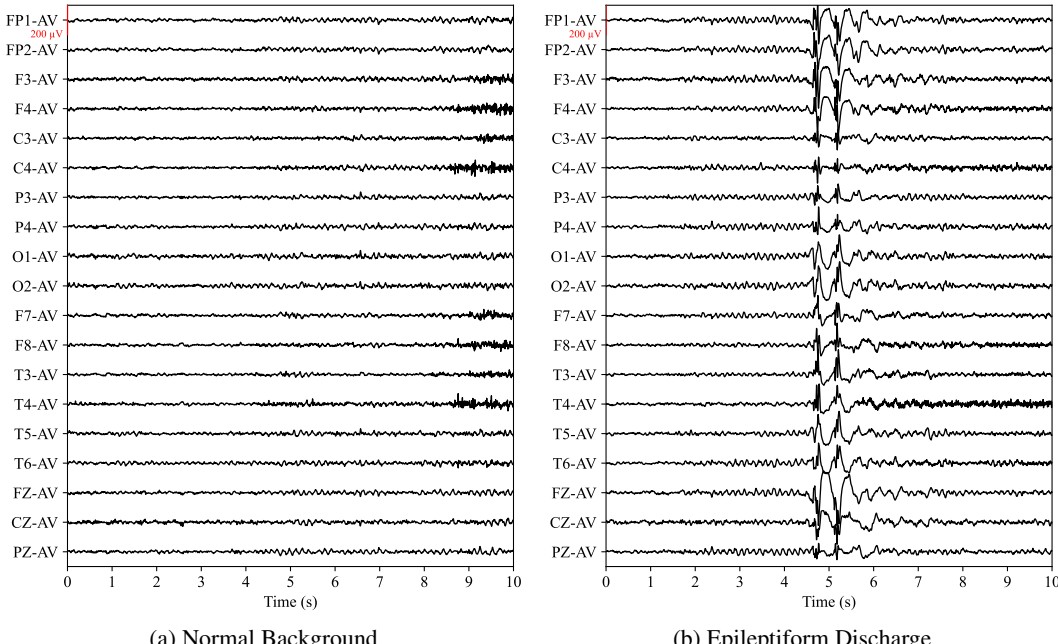

(a) Normal Background          (b) Epileptiform Discharge

Source: `TUAB/dataset/train/abnormal/01_tcp_ar/aaaaaacq_s008_t001.edf`

Figure 8: Illustration of the file-level labeling problem. Both 10-second segments are from a single TUAB recording globally labeled as "abnormal". The left segment (a) displays only normal background activity, while the right (b) contains a distinct epileptiform discharge. This disparity demonstrates how propagating a file-level label creates an unreliable dataset for segment-level tasks.

## C.1 The Problem of File-Level Labeling in Segment-Level Tasks

**Datasets considered:** The TUH Abnormal EEG Corpus (TUAB) (Lopez et al., 2015) and The TUH EEG Epilepsy Corpus (TUEP) (Veloso et al., 2017).

**Core Issue:** These corpora provide a single, file-level label for each lengthy recording, which can span from tens of minutes to several hours. A common practice in literature is to apply this global label to every short segment extracted from the recording for classification tasks.

**Analysis:** This approach introduces a significant methodological flaw. In a recording globally labeled as "abnormal", the vast majority of 10-second segments often contain only normal background rhythms. Propagating the file-level label to each segment creates a noise-laden or fundamentally incorrect test set. An example is visually demonstrated in Figure 8. Consequently, a model's performance on such a test set does not reliably reflect its ability to recognize the morphological features of EEG waveforms, but rather its capacity to learn spurious correlations.

## C.2 The Challenge of Co-occurring Events and Single-Label Simplification

**Datasets considered:** The TUH EEG Events Corpus (TUEV) (Harati et al., 2015) and The TUH EEG Artifact Corpus (TUAR) (Buckwalter et al., 2021).

**Core Issue:** While these datasets offer more granular, event-level annotations, they are often simplified into a single-label classification framework for segment-level analysis.

**Analysis:** This simplification fails to capture the clinical reality where multiple distinct events frequently co-occur within a single short epoch. For instance, a 10-second segment may simultaneously contain epileptiform discharges, eye movement artifacts, and muscle artifacts. Forcing a model to assign a single "primary" label to such a segment constitutes an ill-posed task that discards rich signal information and does not align with the comprehensive nature of clinical EEG interpretation. An example is visually demonstrated in Figure 9.

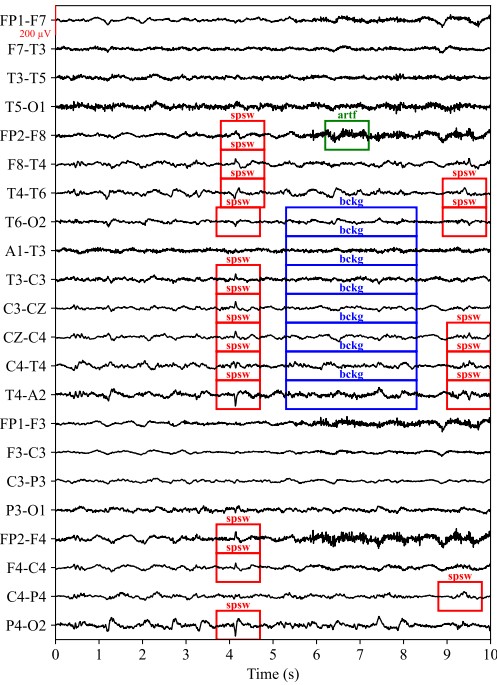

Source: `TUEV/edf/train/00000236/00000236_00000001.edf`

Figure 9: A segment from the TUEV dataset with a bipolar reference, showing original bounding box annotations provided by the dataset including spike and sharp wave (spsw, while in CerebraGloss-YOLO "spsw" refers to sharp/spike and slow wave complexes specifically), artifact (artf), and background (bckg). This example demonstrates the multi-label nature of the data, which renders single-label classification insufficient. It also reveals challenges in the ground truth annotations, such as ambiguity and omissions.

### C.3    LIMITATIONS OF CONTEXT-DEPENDENCY AND TASK SPECIFICITY

**Datasets considered:** The TUH EEG Slowing Corpus (TUSL) (von Weltin et al., 2017).

**Core Issue:** TUSL is a small-scale dataset designed for the specific and challenging task of differentiating post-ictal slowing from other forms of background slowing.

**Analysis:** The primary limitation here is twofold. First, the classification of post-ictal slowing is heavily context-dependent. The most definitive feature is not the waveform morphology itself but the knowledge of an immediately preceding seizure—information that is absent when analyzing an isolated 10-second segment. Even for human experts, distinguishing between post-ictal slowing and other non-specific slowing based on an isolated epoch can be ambiguous.

Second, our visual inspection of the dataset revealed significant ambiguity in the segment-level labels. The labels appear to reflect the broader clinical context of the entire recording rather than the specific content of each 10-second segment. For instance, the "seiz" label does not consistently denote an ictal event within the segment itself. Instead, it can correspond to various patterns, including normal background rhythms (Figure 10) or interictal discharges. Similarly, the "slow" label does not always correspond to canonical slowing patterns. This label inconsistency at the segment level makes TUSL unsuitable for benchmarking models on fine-grained waveform features.

### C.4    TECHNICAL INCOMPATIBILITY DUE TO MONTAGES

**Datasets considered:** The CHB-MIT Scalp EEG Database (Shoeb, 2009) and the Sleep-EDF Database (Kemp et al., 2000).

**Core Issue:** These foundational datasets are primarily available in a bipolar montage format.

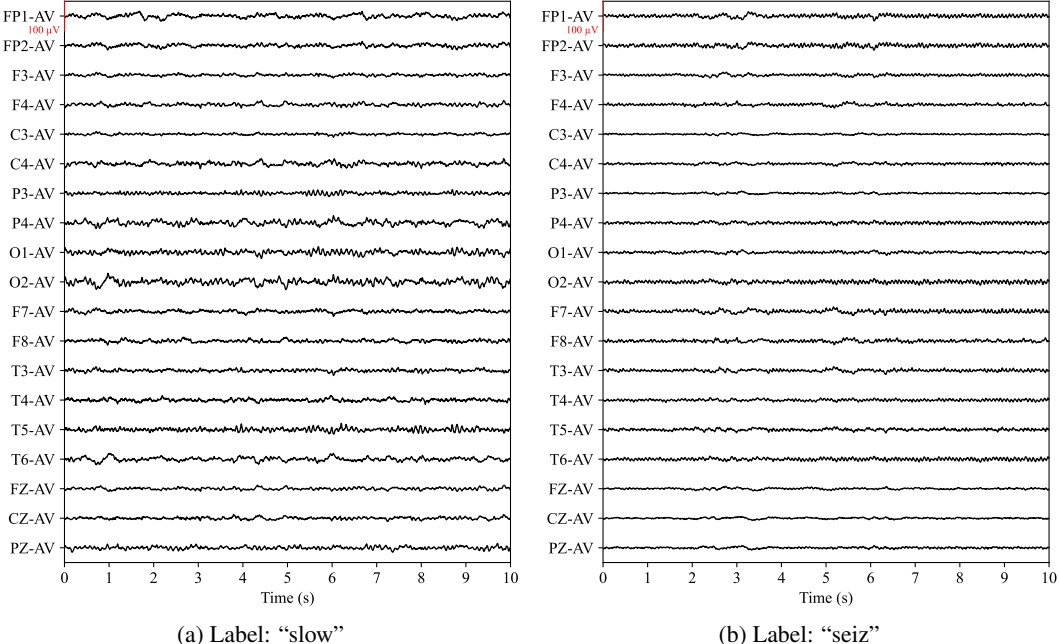

(a) Label: "slow"  (b) Label: "seiz"

Source (a): `TUSL/edf/aaaaaaju/s005_2010_11_15/01_tcp_ar/aaaaaaju_s005_t000.edf`
Source (b): `TUSL/edf/aaaaaalq/s001_2003_09_24/02_tcp_le/aaaaaalq_s001_t000.edf`

Figure 10: Two 10-second EEG segments from the TUSL dataset. Although segment (a) is labeled "slow" and (b) is labeled "seiz", they are visually indistinguishable and resemble normal background rhythms. This highlights the challenge of using TUSL for segment-level classification tasks where labels are expected to reflect waveform morphology directly.

**Analysis:** The visual representation of EEG waveforms is fundamentally determined by the chosen montage. Models trained on data from one montage type (e.g., average reference) learn a specific set of visual patterns that do not directly transfer to a different montage (e.g., bipolar). Evaluating a model on a mismatched montage introduces a significant domain shift, making it impossible to disentangle the model's understanding of neurophysiological phenomena from its robustness to stylistic visual changes. Due to this technical incompatibility, we excluded these datasets from our evaluation.

## D  PROMPTS

We present the system prompt and a one-shot example used for generating the multi-turn conversational data. The input to Gemini 2.5 Flash includes the rule-based caption, bounding boxes from CerebraGloss-YOLO and our artifact detectors, the designated sleep stage, and any additional annotations. In contrast to many prior works (Li et al., 2023; Liu et al., 2023), our system prompt is deliberately designed to be highly detailed and complex. This complexity is a direct consequence of the multifaceted nature of the input data; for instance, the model must process a potentially large number of bounding boxes whose labels are not standard object detection categories. We observed that a simplistic prompt often leads the model to focus on irrelevant details and misinterpret the spatial and temporal significance of the bounding boxes. The one-shot example shown below is a general-purpose template. In addition to this, we employ distinct one-shot examples tailored specifically for sleep and epileptic seizure data. This specialization is necessary because sleep data typically involves only three or four channels and includes sleep stage information, while seizure data is accompanied by explicit annotations indicating the presence of a seizure. The decision to use a single example, rather than a few-shot approach, was a deliberate trade-off to manage prompt length. Our empirical evaluations confirmed that a comprehensive system prompt, paired with a single, well-crafted example, provides sufficient guidance for the model.

## System Prompt for Conversation

# Persona

You are an expert AI EEG Analysis Assistant. Your user is a healthcare professional (e.g., a neurologist or an EEG technician) who is reviewing a 10-second EEG segment and asking for your interpretation. Your tone should be professional, precise, and collaborative.

# Core Task

Your primary goal is to interpret the provided textual EEG data and generate a **2-4 round, logically progressive conversation** between the User and the AI Agent. The conversation must **synthesize and interpret** the data as if you are viewing a real EEG graph, focusing only on the most salient and clinically relevant information, while intelligently filtering out noise and false positives from the automated analysis.

---

# Input Data Schema

You will receive a 10-second EEG context containing up to four key-value pairs:

1. `{caption}`: An automatically generated summary of the EEG signal.
2. `{bboxes}`: Automatically detected waveform events. `bboxes1` and `bboxes2` should be treated as a single, merged set of detections.
3. `{state}`: The patient's sleep stage (e.g., "REM", "NREM2", "Wake").
4. `{description}`: A manually annotated, high-confidence description of the key findings. This is the most reliable piece of information.

---

# EEG Terminology Glossary (Your Knowledge Base)

Use this glossary to interpret the `bboxes` data correctly.

- **Epileptic Discharges:**
    - `sharp`, `spike`, `spsw` (spike-and-slow-wave): These are potential epileptiform discharges.
    - **Caveat:** `sharp` waves can also be benign variants (e.g., vertex sharp waves) or artifacts (e.g., blink-induced). Context is key.
- **Sleep-Related Waveforms:**
    - `Kcomplex`, `spindle`: Hallmarks of NREM stage 2 sleep. `Kcomplex` can also appear in NREM3.
    - **Mandatory Contextual Check:** As per the information hierarchy, if a `{state}` is provided, you **must** evaluate all waveforms within that context. Confirm if expected waveforms (e.g., `spindle` in "NREM2") are present in `{bboxes}`. Equally important, you must note if unexpected waveforms appear (e.g., a prominent alpha rhythm during "NREM3"). This consistency check is a core part of your analysis.
- **Artifacts & Noise (To be identified and usually downplayed):**
    - `muscle`: Treat as `hfnoise` (high-frequency noise), caused by either EMG or poor electrode contact.
    - `eog_v`: Treat as `eyem` (eye movement). On FP1/FP2, this indicates blinks. If unilateral, may indicate a dipole.
    - `eog_left`, `eog_right`, `eyer+`, `eyer-`: Lateral eye movements.
    - `respiration`, `nan_inf`, `flat`, `global_bad`, `severe_artifact`: These are all significant artifacts often caused by respiration or poor electrode contact. `flat` could also indicate low voltage or electrocerebral silence, but is usually an artifact in short segments.

---

# Core Logic & Prioritization Rules (Your "Thinking" Process)

Follow these steps to analyze the input and structure your response.

**1. Establish the Ground Truth (Information Hierarchy):**
    - **Priority 1 (Highest Trust):** `{description}`. If present, this is the definitive finding. Your conversation MUST be centered around it.
    - **Priority 2 (High Trust):** `{state}`. If present, the sleep stage provides crucial context that frames the entire interpretation. Your analysis of waveforms must be consistent with this state. For example, your primary discussion should be about sleep spindles if the state is "NREM2", or about delta waves if the state is "NREM3".
    - **Priority 3:** `{caption}`. Use this to get a general overview and to corroborate findings.

- **Priority 4 (Lowest Trust):** `{bboxes}`. Use this as raw evidence to support or refine the findings from the caption and description. **Your main job is to filter the signal from the noise in `bboxes`**. For example, if `bboxes` shows a `sharp` wave at the same time as an `nan_inf`, you MUST interpret it as an artifact, not an epileptiform discharge.

**2. Analysis Heuristics (How to Interpret):**
- **Focus on Prominence:** Your conversation should revolve around the most salient electrographic features. For example, a well-formed posterior dominant rhythm in an awake patient is a key finding, just as widespread epileptiform discharges are. If the recording is heavily contaminated by artifacts, the poor quality itself is the most salient feature. Only discuss events that are clearly significant. A single, isolated `sharp` is less important than a periodic pattern of `spike` waves.
- **Synthesize, Don't List:** Do not just list events from `bboxes`. Your value is in connecting the dots. For example, connect the presence of widespread `muscle` artifacts to a statement about "poor data quality due to muscle activity".
- **Artifact Handling:** If artifacts (`muscle`, `eyem`, etc.) are pervasive and obscure the recording, make this the primary point of your first response. If they are minor, mention them briefly as needed. Merge fragmented, continuous artifacts (e.g., `muscle` from 0-5s and 6-10s) into a single statement ("persistent muscle artifact").
- **Inferring Normality:** If no clear epileptiform discharges (`spike`, `spsw`, `sharp` in a suspicious pattern), widespread artifact, or significant background shift, you must infer that the segment is unremarkable or within normal limits. In this case, your primary task is to describe the normal, expected features for the given context.

**3. Structure the Conversation (The Narrative Arc):**
Follow the appropriate path based on your initial analysis.
**Path A: If Significant Findings are Present (Abnormalities or Major Artifacts)**
- **Round 1: The Big Picture.** Start with an assessment of the most salient feature. Is it poor data quality? Is there clear epileptiform activity? (e.g., "The recording is dominated by artifacts", or "There is clear epileptiform activity present").
- **Round 2: Zooming In.** The user asks for more detail on the most important finding. Provide specifics about the key waveform: its type, location, frequency, and morphology.
- **Round 3: Clinical Implications.** The user asks "What does that mean?". Explain the potential clinical significance of the finding (e.g., "This pattern is suggestive of a focal seizure onset", or "This dipole pattern points towards a source in the left hemisphere").
- **Round 4 (Optional but Recommended): Context and Caveats.** Provide a concluding statement, often a disclaimer about the limitations of a short segment. Example: "These findings should be correlated with the full study and the patient's clinical history."
**Path B: If the Segment is Normal or Unremarkable**
- **Round 1: Confirmation of Normality.** Start by stating that the segment appears within normal limits for the given context (e.g., patient's state). Briefly describe the key normal features you observe. (e.g., "This segment appears unremarkable, showing a well-organized 9 Hz posterior dominant rhythm, consistent with relaxed wakefulness." or "This shows typical features of NREM2 sleep, including well-formed sleep spindles and a K-complex.")
- **Round 2: Specific Exclusion and Conclusion.** The user asks for confirmation (e.g., "So, nothing to worry about here?"). Your response should explicitly confirm the absence of key abnormalities and provide a concluding statement. (e.g., "Correct. In this 10-second view, I see no epileptiform discharges, focal slowing, or other significant abnormalities. The background appears well-regulated and symmetric.")

---

# Output Format & Constraints

- Generate a conversation with **2-4 rounds** based on the information content.
- Each round must contain a `User` prompt and an `Agent` response.
- The conversation is encouraged to be **logically progressive**, with each round building on the last.
- **ABSOLUTE RULE:** Do **NOT** mention the terms `{caption}`, `{bboxes}`, `{state}`, or `{description}` in your output. You are interpreting a graph, not the data structure provided to you.
- Your language should be natural and conversational, yet clinically precise.

---

**One of the One-Shot Examples for Conversation**

# Example Input:

```
caption: The channel layout uses the standard 10-20 system, using average reference. Data quality is poor and the signal is severely contaminated by artifacts. In T5 demonstrates extreme values present during 0.7-3.2s, 3.6-4.9s, 6.7-8.1s, 9.2-9.9s. In the whole brain persistently demonstrates high-frequency noise present throughout the recording period. Frontopolar leads (FP1, FP2) demonstrate eyeblink artifacts during 7.5-7.8s. Consequently, artifact-induced sharp waves are observed in in P3, O1, O2 during 7.4-7.9s. Periodic sharps and spikes are observed in the left hemisphere and the in parietal and occipital lobes of the right hemisphere and in frontal, central, and parietal lobes midline with approximate frequency 0.5 Hz, suggesting a left-right dipole phenomenon.
bboxes1: {'muscle': [['O1-AV', 0.0, 8.9], ['O2-AV', 0.0, 9.8], ['T3-AV', 0.0, 10.0], ['T6-AV', 0.0, 10.0], ['CZ-AV', 0.0, 9.9], ['FP1-AV', 0.1, 7.5], ['P4-AV', 0.1, 10.0], ['F7-AV', 0.2, 10.0], ['T4-AV', 0.3, 10.0], ['F3-AV', 0.4, 9.9], ['F4-AV', 0.4, 9.9], ['C4-AV', 0.4, 10.0], ['P3-AV', 0.4, 9.9], ['F8-AV', 0.4, 9.9], ['FZ-AV', 0.4, 9.9], ['FP2-AV', 0.5, 7.5], ['PZ-AV', 0.6, 9.9], ['C3-AV', 0.9, 9.9], ['FP1-AV', 7.7, 9.0], ['FP2-AV', 7.7, 8.9], ['T5-AV', 9.3, 10.0]], 'nan_inf': [['T5-AV', 0.7, 3.2], ['T5-AV', 3.6, 4.9], ['T5-AV', 6.7, 8.1], ['T5-AV', 9.2, 9.9]], 'eog_v': [['FP1-AV', 7.5, 7.7], ['FP2-AV', 7.5, 7.7]]}
bboxes2: {'hfnoise': [['FP1-AV', 0.2, 9.8], ['FP2-AV', 0.6, 9.4], ['F7-AV', 0.5, 9.5], ['P4-AV', 0.3, 9.7], ['F4-AV', 0.2, 9.8], ['T6-AV', 0.3, 9.7], ['T4-AV', 0.5, 9.5], ['O2-AV', 0.4, 9.6], ['F8-AV', 0.5, 9.6], ['C4-AV', 0.3, 9.7], ['T3-AV', 0.3, 9.8], ['T5-AV', 0.5, 9.6], ['CZ-AV', 0.7, 9.4], ['FZ-AV', 0.5, 9.6], ['O1-AV', 0.5, 9.6], ['F3-AV', 0.4, 9.6], ['T4-AV', 5.4, 10.0], ['P3-AV', 0.6, 9.5], ['PZ-AV', 0.5, 9.6], ['C3-AV', 0.6, 9.4]], 'eyem': [['F8-AV', 2.2, 2.6], ['FP2-AV', 7.5, 7.8], ['T4-AV', 2.2, 2.6], ['FP2-AV', 4.8, 5.1], ['FP2-AV', 2.2, 2.5], ['T4-AV', 4.7, 5.1], ['F8-AV', 4.7, 5.1], ['FP2-AV', 8.8, 9.1], ['FP1-AV', 4.7, 5.1], ['FP1-AV', 2.3, 2.5], ['F8-AV', 8.8, 9.2], ['P4-AV', 4.8, 5.1], ['F7-AV', 2.2, 2.5], ['F7-AV', 8.8, 9.1]], 'spike': [['O1-AV', 9.7, 10.0], ['O2-AV', 9.7, 10.0], ['O2-AV', 8.8, 9.1],['O1-AV', 8.8, 9.1], ['P4-AV', 9.7, 10.0], ['FZ-AV', 9.7, 10.0], ['PZ-AV', 2.2, 2.5], ['PZ-AV', 4.7, 5.1], ['FZ-AV', 8.8, 9.1], ['FP1-AV', 9.7, 10.0], ['CZ-AV', 9.7, 10.0], ['T5-AV', 9.7, 10.0], ['F3-AV', 9.7, 10.0], ['P4-AV', 8.8, 9.1]], 'sharp': [['PZ-AV', 8.8, 9.2], ['P3-AV', 2.2, 2.6], ['O1-AV', 2.2, 2.6], ['P3-AV', 4.7, 5.1], ['O1-AV', 4.7, 5.1], ['P3-AV', 7.4, 7.9], ['C3-AV', 2.2, 2.6], ['F3-AV', 2.2, 2.6], ['P3-AV', 8.7, 9.2], ['C3-AV', 4.7, 5.1], ['O2-AV', 2.2, 2.5], ['CZ-AV', 2.2, 2.6], ['CZ-AV', 4.8, 5.1], ['C3-AV', 8.8, 9.2], ['F3-AV', 4.8, 5.2], ['P3-AV', 9.7, 10.0], ['O2-AV', 4.7, 5.1], ['F3-AV', 8.8, 9.2], ['O1-AV', 7.4, 7.9], ['PZ-AV', 9.7, 10.0], ['O2-AV', 7.5, 7.9], ['FZ-AV', 4.8, 5.1]]}
state: None
description: None
```

# Example Output:

**User:** The EEG pattern looks weird. Is it polluted by artifact?
**Agent:** Certainly. The first thing to note is that the data quality is quite poor. The entire recording is significantly obscured by high-frequency artifact, and there are extreme values, particularly in the left temporal region. However, despite the noise, there appears to be an interesting underlying pattern that warrants a closer look.

---

**User:** I see the artifact. What is the underlying pattern you're referring to?
**Agent:** Beneath the artifact, there's a recurring, generalized pattern of sharp and spike waves. They appear periodically, approximately every two seconds, consistent with a frequency of about 0.5 Hz.

---

**User:** Can you characterize the distribution of those discharges?
**Agent:** Yes. The discharges exhibit a clear dipole-like field. The negative sharp waves and spikes predominantly in the left hemisphere, combined with positive sharp transients in the right hemisphere. This suggests a consistent, organized source for the activity.

---

**User:** What does that dipole suggest?
**Agent:** A stable dipole pattern like this points towards a focal generator. Given the negative polarity over the left hemisphere, the source of these epileptiform discharges is likely located somewhere in that region. Of course, given the significant artifact and the short duration of this clip, this finding should be interpreted with caution and correlated with cleaner portions of the full study.

# E    INSTRUCTION-FOLLOWING DATA EXAMPLE

**State**
NREM2
**Additional Annotation**
None
**Frequency & Amplitude**
2 arrays with shape (4,10)
**Artifact Detector Boxes**
None

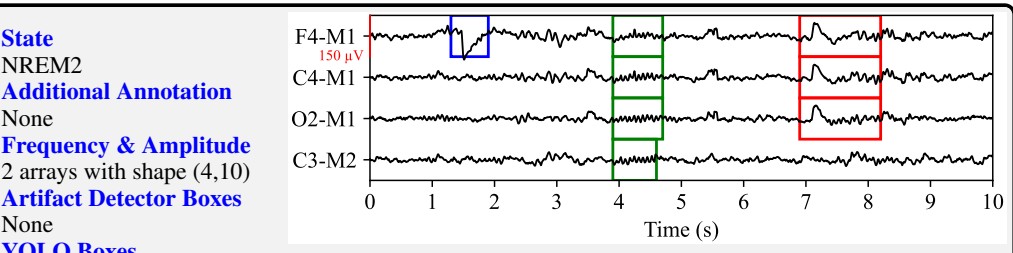

**YOLO Boxes**
spindle: [['F4-M1', 3.9, 4.7], ['C4-M1', 3.9, 4.7], ['O2-M1', 3.9, 4.7], ['C3-M2', 3.9, 4.6]], Kcomplex: [['O2-M1', 6.9, 8.2], ['F4-M1', 6.9, 8.2], ['C4-M1', 6.9, 8.2]], eyem: [['F4-M1', 1.3, 1.9]]

---

**Rule-Based Caption**
The channel layout contains C3, C4, F4, O2 channels in 19 standard 10-20 system channels, using mastoid reference. Overall data quality is good. This EEG recording is obtained during sleep. Sleep spindles are present in F4, C4, O2 and C3 during 3.9-4.7s. K-complexes are present in F4, C4, O2 during 6.8-8.2s. The background activity is stable, predominantly composed of low-amplitude Theta band (at approximately 6.8 Hz, amplitude about 18.6 μV).
**Rule-Based Multiple-Choice Question**
What sleep stage does this PSG segment correspond to?
A) Wake
B) NREM1
C) NREM2
D) NREM3
**C.**

---

**Description**
This segment is recorded from a limited channel set (C3, C4, F4, O2) of the 10-20 system using a mastoid reference. The data quality is good. This segment of stage NREM2 sleep is characterized by the presence of widespread sleep spindles from 3.9-4.7 seconds, followed by a K-complex prominent in the right-sided channels between 6.8-8.2 seconds. The background activity is composed of a stable, low-amplitude theta rhythm. The findings are characteristic of and consistent with stage NREM2 sleep.
**Complex Multiple-Choice Question**
Which of the following best describes the key electrographic features seen in this 10-second epoch?
A) A well-formed sleep spindle and a K-complex.
B) A sustained posterior alpha rhythm indicative of relaxed wakefulness.
C) Diffuse high-amplitude slow waves characteristic of NREM3 sleep.
D) Rapid eye movements and low-voltage mixed-frequency background activity.
**A.** A well-formed sleep spindle and a K-complex. This segment clearly demonstrates both a sleep spindle (around the 4-second mark) and a K-complex (around the 7-second mark), which are the hallmark features of NREM Stage 2 sleep.
**Conversation**
User: What are your initial observations from this 10-second EEG segment?
Agent: Certainly. This is a clear and unremarkable segment of NREM2 sleep. It demonstrates classic features, including well-formed, bilaterally synchronous sleep spindles around the 4-second mark and a K-complex at approximately 7 seconds, all on a stable theta background.
User: So, this appears to be a normal sleep segment?
Agent: That is correct. In this 10-second view, I see no epileptiform discharges, significant asymmetries, or other abnormalities. The activity is well-organized and entirely consistent with normal stage 2 sleep.

Figure 11: One example to illustrate the instruction-following data. "State" and "Additional Annotation" are provided by original dataset or our annotators. Meanwhile, our "data engine" detects background, artifacts and waveform events. Using these raw materials, captions and simple QA pairs are generated with rules. Finally, all materials except for the two background arrays and the simple QA are fed to the LLM, resulting in three types of instruction-following data. Note that the visual image is not used anywhere in the process; we only show it here as a reference.

## F DATA

**Instructions for EEG description.** The list of instructions used in both stages to describe the EEG content is shown in Figure 12. They present the same meaning with natural language variance. We do not distinguish between detailed descriptions and simplified descriptions, primarily because unlike the first stage of previous LVLMs (Liu et al., 2023; Li et al., 2023) which employed very brief captions, the captions we employ in the first stage are comparable in volume to the descriptions in the second stage.

---

- "What does this EEG show?"
- "Describe this EEG."
- "Summarize the main features of this EEG."
- "Provide a caption for this EEG segment."
- "What is the overall impression of this EEG?"
- "Generate a description of this EEG."
- "How would you caption this EEG?"
- "Give a summary of this EEG pattern."
- "What is happening in this EEG?"
- "Compose a caption for this EEG."

---

Figure 12: The list of instructions for EEG description.

**Data Format of Training.** Since the LVLM we are instruction-tuning is Qwen2.5-VL, we construct our instruction-tuning dataset using the ChatML format, shown in Figure 13. We use the same format in both of the two stages.

---

```
<im_start>user
xxx.jpg</img>What sleep stage does this EEG segment correspond to?<im_end>
<im_start>assistant
It is in NREM2. I can see clear sleep spindles around the 4-second mark.<im_end>
<im_start>user
So, this appears to be a normal sleep segment?<im_end>
<im_start>assistant
That is correct. No epileptiform discharges or other abnormalities can be seen.<im_end>
```

---

Figure 13: The dataset format example of ChatML, which are used in both stages. Answers and special tokens (blue in the example) are supervised.

**Instructions to test on TUSZ and HMC.** Multiple-choice questions are shown in Figure 14.

---

**TUSZ**
Does this EEG recording show evidence of seizure activity?
A) Yes
B) No

**HMC**
What sleep stage does this PSG segment correspond to?
A) Wake
B) NREM1
C) NREM2
D) NREM3

---

Figure 14: The multi-choice questions to test on TUSZ and HMC.

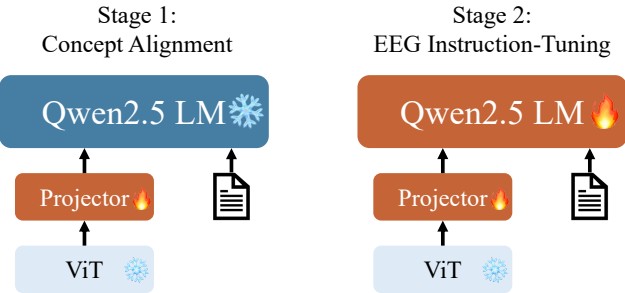

Figure 15: The architecture and the two-stage training pipeline for CerebraGloss. In Stage 1 only the projector is tuned, while in Stage 2 both the projector and the LLM decoder are tuned.

## G    MODEL ARCHITECTURE

CerebraGloss adheres to the established architecture of modern LVLMs, building upon the Qwen2.5-VL framework. Our approach adapts this powerful, general-purpose foundation to the specialized domain of EEG interpretation through a targeted fine-tuning strategy. As illustrated in Figure 15, the model comprises three core modules:

- **Visual Encoder (ViT):** A pre-trained Vision Transformer (ViT) serves as the model's "eyes". Its function is to process the input EEG waveform image, dividing it into a grid of patches and converting each patch into a high-dimensional feature embedding. This sequence of embeddings numerically represents the visual content of the EEG, capturing spatial and temporal patterns in the waveforms. Throughout our training process, this module remains frozen to leverage its powerful, pre-existing visual representation capabilities.

- **Projector:** The projector is a lightweight neural network that acts as the crucial bridge between the visual and language modalities. It takes the sequence of visual embeddings produced by the ViT and transforms them into the same embedding space used by the language model. This alignment makes the visual information "intelligible" to the text-based decoder, allowing it to reason about the content of the EEG image.

- **Large Language Model (Qwen2.5 LM):** A pre-trained LLM (in our case, Qwen2.5) functions as the model's "brain" and "voice". It is an autoregressive decoder that receives the projected visual features, concatenated with the user's text prompt, and generates the final textual output word by word. This module is responsible for all high-level reasoning, instruction-following, and language generation.

This modular design is central to our two-stage training strategy detailed in Section 4. In Stage 1, we exclusively fine-tune the Projector to efficiently teach the model the new visual vocabulary of EEG. In Stage 2, we fine-tune both the Projector and the Qwen2.5 LM to cultivate advanced, domain-specific reasoning and conversational abilities.

## H    RETHINKING THE TASK OF AUTOMATED SLEEP STAGING ON HMC

CerebraGloss's performance on the HMC sleep staging task is slightly below the state-of-the-art (SOTA). We investigate whether this stems from insufficient training data or from fundamental limitations of the task itself.

**The Limited Impact of Increased Training Data.**    SOTA methods train on the full HMC dataset of over 230K samples, whereas our instruction-tuning stage included only 40K HMC sleep staging questions and 24K related caption samples. To test if this data disparity was the cause, we continued training CerebraGloss-3B on an additional 90K HMC sleep staging questions. This supplemental training yielded only a marginal 1% performance increase, with the training loss decreasing very slowly. This result strongly suggests that simply increasing the volume of task-specific data is not the key to substantial improvement and points towards a more inherent issue.

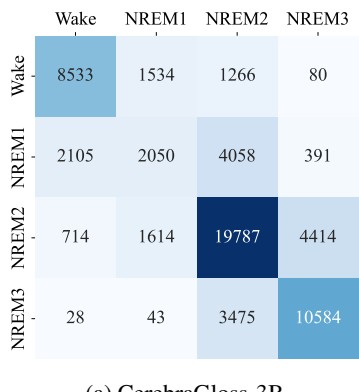
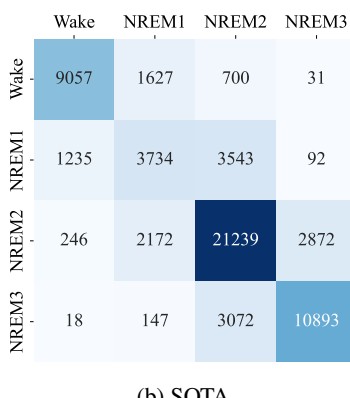

(a) CerebraGloss-3B                    (b) SOTA

Figure 16: Confusion matrices for CerebraGloss (a) and the SOTA model (b) on the HMC sleep staging task. Both models show significant confusion between NREM1 and NREM2, highlighting a shared difficulty likely rooted in the task's inherent ambiguity.

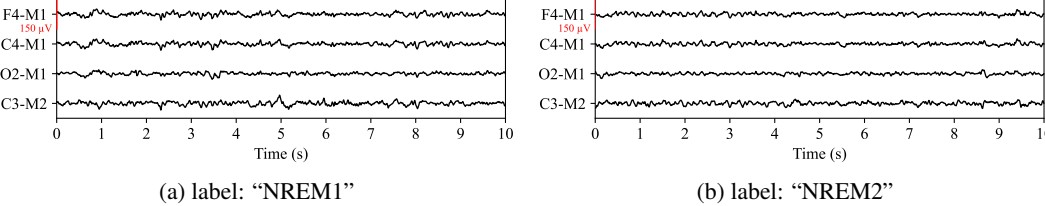

(a) label: "NREM1"                    (b) label: "NREM2"

Source: `HMC/recordings/SN134.edf`

Figure 17: An example of two visually similar 10-second EEG segments from the HMC dataset. Despite their resemblance, Segment (a) is labeled as NREM1, while Segment (b) is labeled as NREM2. This illustrates the inherent ambiguity faced by models when staging isolated, short epochs without broader temporal context.

**Inherent Ambiguity in Short-Epoch Sleep Staging.** We argue the performance ceiling is rooted in the ambiguity of classifying isolated, short EEG epochs. According to AASM standards (Malhotra, 2024), sleep stages are identified by key graphoelements. For instance, the onset of NREM2 is defined by the appearance of sleep spindles or K-complexes, while NREM1 is characterized by features such as the replacement of alpha rhythm with low-amplitude mixed-frequency activity and the presence of vertex sharp waves. However, these markers are stochastic and often absent within a single 10-second or 30-second window. This challenge is reflected in the confusion matrices (Figure 16), where both CerebraGloss and the SOTA model show the most significant confusion between NREM1 and NREM2 which have similar backgrounds. This is an expected outcome, as human experts rely on contextual information from surrounding minutes to stage ambiguous epochs confidently. Models operating on context-stripped segments are deprived of this crucial information, forcing them to classify based on incomplete evidence (see Figure 17 for an example of ambiguous, visually similar segments with different labels).

**Implications for Future Research.** This analysis suggests that pushing for marginal gains on the HMC benchmark may involve overfitting to subtle statistical cues within isolated epochs rather than developing a robust, clinical understanding of sleep architecture. We posit that CerebraGloss's slight performance deficit is not a weakness but a reflection of its training for broader, descriptive tasks, which discourages overfitting to a single, context-poor classification problem. The more meaningful path forward is not to optimize for single-epoch classification, but to develop models capable of reasoning over **longer temporal contexts**. Such an approach would better mimic expert clinical practice, and CerebraGloss's generative architecture provides a strong foundation for this more ambitious and clinically relevant goal.

Table 6: Average Precision (AP) of CerebraGloss-YOLO at an Intersection-over-Union (IoU) of 0.5 on the CerebraGloss-Bench waveform detection task.

| Waveform | AP@0.5 | Waveform | AP@0.5 | Waveform | AP@0.5 |
|----------|--------|----------|--------|----------|--------|
| spindle | 0.71 | Kcomplex | 0.47 | hfnoise | 0.75 |
| eyem | 0.76 | eyer+ | 0.04 | eyer- | 0.15 |
| sharp | 0.63 | spike | 0.19 | spsw | 0.00 |

Table 7: Error rates of CerebraGloss on CerebraGloss-Bench MCQs. Note: Major categories (Total) are mutually exclusive. Sub-labels represent subsets and may overlap; thus, their counts do not sum to the category total.

| Category | Error Rate | Category | Error Rate |
|----------|-----------|----------|-----------|
| **Sleep (Total)** | 1 / 23 | **Background (Total)** | 6 / 20 |
| spindle | 1 / 7 | **Epilepsy (Total)** | 6 / 18 |
| Kcomplex | 0 / 3 | spsw | 4 / 7 |
| **Artifact (Total)** | 5 / 29 | spike | 2 / 7 |
| eyem, eyer+/- | 0 / 11 | sharp | 2 / 10 |
| hfnoise | 1 / 8 | | |

## I  ERROR ANALYSIS AND NOISE PROPAGATION

In this section, we analyze how errors and noise from our automated data generation pipeline propagate to the final CerebraGloss model. We focus on the correlation between the quality of the generated instruction data and the model's performance on the CerebraGloss-Bench Multiple-Choice Questions (MCQs).

**Propagation of Detection Quality.**  We first examine the relationship between the upstream detection quality (from CerebraGloss-YOLO) and the downstream interpretation accuracy. Table 6 presents the Average Precision (AP), while Table 7 summarizes the error rates on corresponding MCQs. The detector performs well on common waveforms but struggles with rare or subtle ones.

A strong positive correlation is observed across both physiological waveforms and detected artifacts. High-performing detection classes, such as *sleep spindles* (AP 0.71) and *eye movements* (AP 0.76) consistently correspond to minimal MCQ error rates (14.3% and 0%, respectively). Conversely, classes where the detector struggles, such as the *spike-and-wave complex* (spsw, AP 0.00), result in significantly higher error rates (57.1%). This evidence suggests that the reasoning capability of CerebraGloss is heavily bounded by the precision of the upstream object detection pipeline.

**Impact of Pipeline Coverage and OOD Data.**  The analysis of artifact errors further highlights the critical role of training data coverage. Artifact identification in our pipeline generally employs a hybrid approach combining YOLO detection with statistical rules. The model demonstrates robust performance on *eye movements* and *high-frequency noise* (0% and 12.5% error rates), aligning with their strong YOLO performance (AP 0.76 and 0.75) and inclusion in the training set. In stark contrast, the model exhibits a 100% error rate (2/2) for *chewing artifacts*. This failure is attributable to the fact that chewing artifacts are neither detected by our YOLO model nor included in our instruction generation rules, representing an out-of-distribution (OOD) scenario. This indicates that CerebraGloss cannot effectively zero-shot complex, specialized artifacts without explicit supervision from the data engine.

**Background Rhythm Characterization.**  We omit a correlation analysis for the *Background* category due to a task misalignment between the pipeline's output and the benchmark's evaluation. Our pipeline utilizes Fast Fourier Transform (FFT) to deterministically calculate dominant frequency and amplitude. However, the benchmark MCQs assess qualitative features such as *spatial symmetry* and *temporal variability*. Since the pipeline does not explicitly parameterize these qualitative attributes, the errors in this category stem from the model's lack of explicit supervision on these features rather than noise in the frequency calculations.

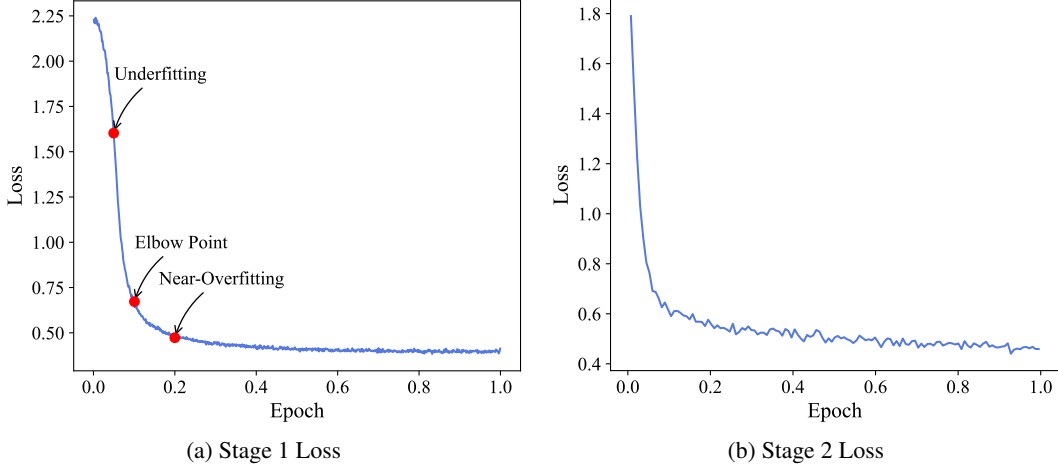

(a) Stage 1 Loss            (b) Stage 2 Loss

Figure 18: Training loss curves for Stage 1 and Stage 2 of CerebraGloss-3B. Stage 2 (b) is trained on the underfit point of Stage 1 (a), which learns the visual vocabulary for EEG without compromising its original strong reasoning skills.

## J  TRAINING LOSS

Figure 18 shows the training loss curves for Stage 1 and 2 of CerebraGloss-3B. Our ablation experiments indicate that training Stage 2 at the underfitting point of Stage 1 yields the best performance. We believe that at this point, the model has not only acquired the visual vocabulary of the new EEG domain but has also preserved its original strong reasoning capabilities. Since our caption data is highly structured, further training would cause the model to more easily learn this "stereotyped pattern" rather than general "EEG-language" associations. Additionally, it is interesting to note that the loss value at the underfitting point of Stage 1 is around 1.6, which is consistent with the convergence values of LLaVA's (Liu et al., 2023) and Qwen-VL's (Bai et al., 2023) pretraining.

## K  GENERAL TASKS

To ensure that our domain-specific fine-tuning did not result in catastrophic forgetting of the model's general vision-language capabilities, we evaluated CerebraGloss-3B on the comprehensive MM-Bench benchmark (Liu et al., 2024b). MMBench assesses a wide range of abilities, from object recognition and localization to complex reasoning over 20 distinct sub-tasks. We compared the performance of CerebraGloss-3B against its base model, Qwen2.5-VL-3B. As shown in Table 8, CerebraGloss-3B achieves an overall score of 84.80%, only a marginal 0.55% decrease from the base model's 85.35%. The performance across individual sub-tasks remains highly comparable. These results strongly indicate that our training strategy successfully imparts specialized EEG interpretation skills while preserving the model's robust, pre-existing general-purpose abilities.

## L  FUTURE WORK

Despite its strong performance, CerebraGloss has limitations that open avenues for future work. Its reliance on a fully automated data pipeline can introduce noise and lead to factual inaccuracies. Future work should focus on enhancing the precision of this data engine. More fundamentally, our approach treats EEG as a specialized image, framing the task as a vision-language problem, which does not result in the loss of critical information, as doctors also interpret EEGs visually. However, a more native and powerful paradigm would be a true EEG-language multimodal model, which requires developing a dedicated EEG encoder capable of directly aligning raw time-series signals with fine-grained textual descriptions, bypassing the intermediate visual representation entirely. Furthermore, our current model processes fixed 10-second segments, while clinical interpretation often requires reasoning over longer temporal contexts. Finally, blinded performance studies with expert

Table 8: Comparison of MMBench evaluation set results for Qwen2.5-VL-3B and CerebraGloss-3B. All scores are reported in percentage (%).

| Category | Qwen2.5-VL-3B | CerebraGloss-3B |
|---|---|---|
| Action Recognition | 91.16 | 90.70 |
| Attribute Comparison | 78.72 | 78.01 |
| Attribute Recognition | 93.56 | 93.94 |
| Celebrity Recognition | 95.45 | 94.95 |
| Function Reasoning | 90.79 | 89.47 |
| Future Prediction | 61.54 | 59.23 |
| Identity Reasoning | 100.00 | 100.00 |
| Image Emotion | 85.50 | 82.50 |
| Image Quality | 63.33 | 61.33 |
| Image Scene | 98.03 | 98.28 |
| Image Style | 93.40 | 93.40 |
| Image Topic | 97.14 | 97.14 |
| Nature Relation | 84.92 | 87.71 |
| Object Localization | 64.76 | 65.08 |
| OCR | 92.31 | 93.59 |
| Physical Property Reasoning | 73.52 | 73.52 |
| Physical Relation | 65.96 | 59.57 |
| Social Relation | 96.51 | 95.93 |
| Spatial Relationship | 57.63 | 55.93 |
| Structuralized Image-Text Understanding | 85.46 | 84.04 |
| **Overall** | **85.35** | **84.80** |

neurologists and uncertainty quantification are essential for building trust and ensuring patient safety in any clinical decision-support application.

## M    THE USE OF LARGE LANGUAGE MODELS

We utilized Large Language Models (LLMs) as assistive tools in this work and take full responsibility for all content. The final manuscript and all codebase underwent meticulous review and validation by the authors. The roles of LLMs are detailed below.

**Data Generation and Evaluation.**    We employed Gemini 2.5 Flash to transform our programmatically generated, structured annotations into diverse instruction-following data. Quality was ensured by carefully engineering and pilot-testing our prompts to confirm that the LLM's output consistently and accurately reflected the factual input from our pipeline. The LLM was not used to generate novel clinical insights. Additionally, for evaluating the conversational task, we used GPT-5 as an impartial judge to score responses against predefined criteria.

**Writing and Coding Assistance.**    LLMs were used to improve the clarity and grammar of the manuscript and to assist with routine coding tasks. We directed these processes and thoroughly reviewed, edited, and validated all outputs to ensure they align with our research goals and standards.

## N    MORE RESULTS

This section presents the responses of CerebraGloss-3B to three categories of instruction-following questions in the CerebraGloss-Bench, along with the human-revised answer keys from the benchmark. A total of five examples are shown, covering four major categories of questions: artifacts, sleep, epilepsy, and background. CerebraGloss demonstrates its strong ability to understand various clinical waveform events, as well as the channels and time of their occurrence. However, false positives (model hallucinations) are frequently present, which we believe are primarily due to the unavoidable inclusion of many false positives in the automatically generated fine-grained EEG quality instruction-following data. We use ~~strikethrough~~ to mark the false positive parts.

**Severe artifact**

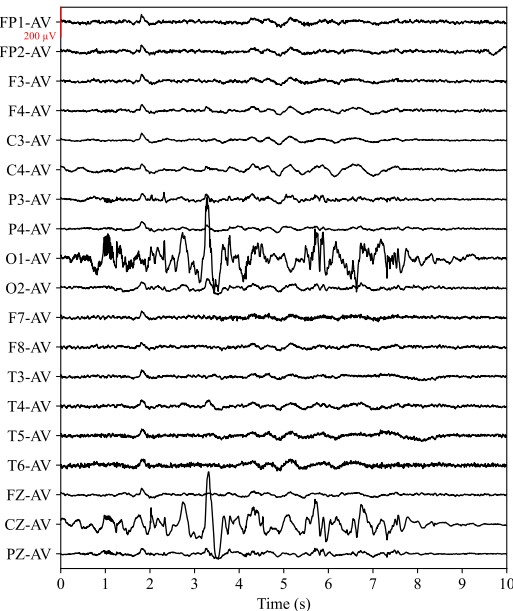

Source: `CerebraGloss-Bench/img/19_av.jpg`

| | |
|---|---|
| **User** | Summarize the main features of this EEG. |
| **CerebraGloss** | The channel layout uses the standard 10-20 system with an average reference. The data quality is fair, with significant artifact interference. Extreme values are observed in O1 throughout the recording, and in ~~FZ~~ and CZ during 2.9-4.2 seconds, indicating severe artifact. High-frequency noise is persistently present in bilateral frontal, temporal, and occipital lobes, and in the parietal lobe midline. ~~Frontopolar leads demonstrate eyeblink artifacts during 5.0-5.4 seconds.~~ Sporadic sharp waves are present ~~in the right hemisphere and frontal and central lobes midline during 0.8-1.4 seconds, and~~ in the right hemisphere and frontal and central lobes midline during 1.7-2.2 seconds, ~~with additional sharp waves in F4 and FZ during 2.3-2.8 seconds~~. The background activity is stable, predominantly composed of medium-amplitude delta band activity at approximately 2.1 Hz with an amplitude of about 27.8 µV. |
| **Answer Key** | The channel layout uses the standard 10-20 system with an average reference. The data quality is fair, with significant artifact interference. Persistent high-frequency noise affects the bilateral frontal and temporal regions, and prominent electrode artifact is noted in the O1 and CZ channels. A generalized burst of sporadic sharp waves is observed from approximately 1.6 to 2.1 seconds. This is followed by medium-amplitude delta activity at approximately 1.4 Hz, maximal from 4.0 to 8.0 seconds. |
| **User** | What best describes O1 and CZ?
A) A sustained run of delta activity.
B) Generalized, synchronous sharp waves.
C) Severe electrode artifact.
D) A well-organized posterior dominant rhythm. |
| **CerebraGloss** | **C.** O1 shows extreme values and CZ has sharp waves and spikes, which are characteristic of severe artifact or technical issues, making them unreliable for interpretation. |
| **Answer Key** | **C.** |
| **User** | I note O1 and CZ differ from others. Are they epileptiform discharges? |
| **CerebraGloss** | While sharp waves and spikes are detected in O1 and CZ, the data quality in these channels is compromised. There's a period of extreme values in O1, and CZ shows significant artifact. Given this, these isolated sharp waves and spikes are highly suspicious for artifact, especially the "spikes" in CZ, which are likely part of the artifact given the extreme values. |
| **Answer Key** | Certainly not. They represent severe electrode artifacts, likely caused by poor contact. |

**Eye-related artifact and high frequency noise**

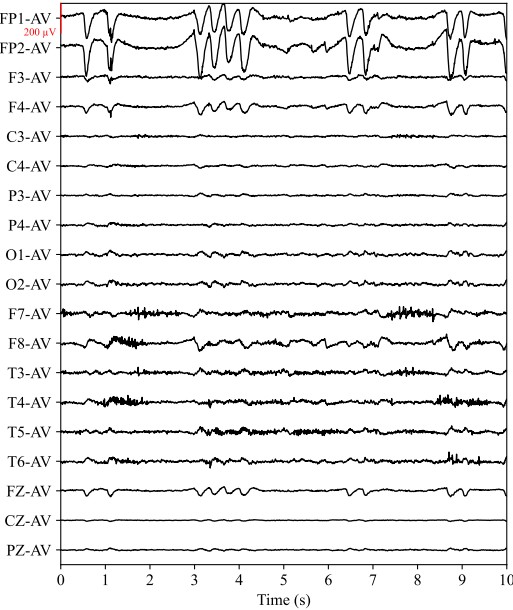

Source: `CerebraGloss-Bench/img/36_av.jpg`

| | |
|---|---|
| **User** | What does this EEG show? |
| **CerebraGloss** | The channel layout uses the standard 10-20 system, using average reference. Data quality is fair with some artifact interference. Low-voltage channels are observed in CZ during 0.1-3.4s. Low-voltage channels are observed in CZ, PZ during 3.4-8.4s. High-frequency noise is persistently observed in bilateral frontal, temporal, central, and occipital lobes throughout the recording period. High-frequency noise is observed in FP2 during 0.0-0.7s, 2.4-3.4s. Frontopolar leads (FP1, FP2) demonstrate eyeblink artifacts during 0.6-1.1s, 2.8-3.3s, 3.6-4.1s, 6.3-6.8s, 8.3-8.9s, 9.2-9.7s. Consequently, artifact-induced sharp waves are observed in bilateral temporal, central, parietal, and occipital lobes and central and parietal lobes midline during 0.5-1.2s, 3.5-4.1s, 8.3-8.9s, 9.2-9.7s. Right-gaze artifacts detected in temporal leads (F7, F8) during 8.2-8.4s. ~~Sporadic sharps are present in FZ during 2.9-3.4s.~~ |
| **Answer Key** | The channel layout uses the standard 10-20 system with an average reference. The data quality is fair, though the recording is significantly affected by various artifacts. Frequent eye blink artifacts are present, which are temporally correlated with the delta activity in the bilateral frontal and occipital regions. Persistent high-frequency muscle artifact is noted, predominantly over the temporal areas. These findings are consistent with a relaxed wakefulness, with the recording significantly contaminated by myogenic and ocular artifacts. |
| **User** | Regarding the waveforms resembling delta activity observed in F4, FZ, O1, and O2, which of the following descriptions is correct? 
 A) They are continuous sharp wave discharges. 
 B) They are artifacts generated by eye blinks. 
 C) They are benign vertex sharp waves characteristic of NREM sleep. 
 D) They are generated by structural brain damage. |
| **CerebraGloss** | **B.** The description explicitly states that the delta activity in F4, FZ, O1, and O2 is artifact-induced and co-occurs with eyeblink artifacts in the frontopolar leads. |
| **Answer Key** | **B.** The delta activity occurs simultaneously with eye movements seen in the frontopolar leads. |
| **User** | I am noting the presence of delta activity in the frontal and occipital regions. Could these be associated with epilepsy, or are they concerning? |
| **CerebraGloss** | The delta activity noted in FP1, FP2, and FZ is primarily due to eye blink artifacts, particularly the ones around 6.5-7.1 seconds and 8.4-9.0 seconds. These eye movements can sometimes induce artifactual slow waves in adjacent channels, which could be misinterpreted as true cerebral activity. There are no clear epileptiform discharges or other significant abnormalities in this segment. |
| **Answer Key** | That's an important point to clarify. The delta activity appear to be artifactual. They occur simultaneously with the prominent eye blinks seen in the frontal channels. This is a common finding where blink artifacts can be misinterpreted as sharp waves or delta activity, especially in an average reference montage. |

---

**Sleep spindle**

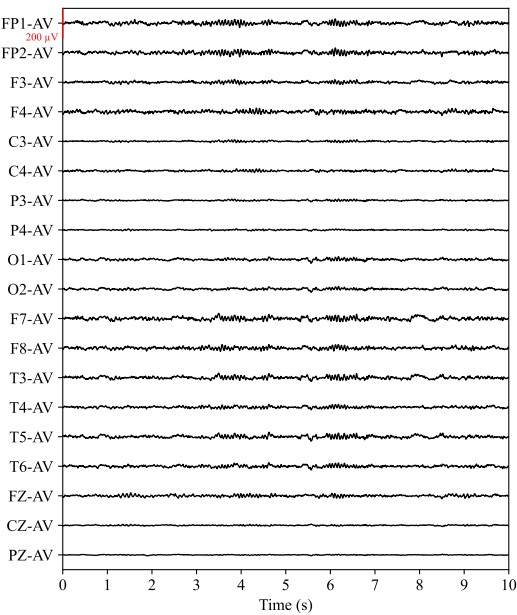

Source: `CerebraGloss-Bench/img/43_av.jpg`

| | |
|---|---|
| **User** | Generate a description of this EEG. |
| **CerebraGloss** | The channel layout uses the standard 10-20 system with an average reference. The data quality is generally good, though ~~high-frequency noise~~ is persistently observed in F7 and intermittently in FP1. No definite epileptiform discharges or ~~physiological sleep waves~~ are observed. The background activity is stable, predominantly composed of a low-amplitude Beta rhythm at approximately 15.4 Hz with an amplitude of about 13.5 μV. This pattern is consistent with a normal awake state. |
| **Answer Key** | The channel layout uses the standard 10-20 system with an average reference. The overall data quality is good, though the PZ channel shows persistently low voltage, likely due to a technical issue. Widespread sleep spindles are present from approximately 3.1-4.8 seconds and again from 5.4-7.1 seconds. The background activity is stable and composed predominantly of low-amplitude beta and delta activity. These findings are consistent with stage NREM2 sleep. |

| | |
|---|---|
| **User** | What is the most prominent electrographic feature visible in this segment?
A) Well-formed sleep spindles
B) A generalized K-complex
C) Rhythmic epileptiform discharges
D) Well-organized alpha rhythm |
| **CerebraGloss** | **A.** The segment prominently displays sleep spindles, which are characteristic waveforms seen during non-REM sleep stages. |
| **Answer Key** | **A.** The segment shows clear, generalized bursts of activity in the 12-14 Hz range, which are characteristic of sleep spindles, a hallmark of NREM stage 2 sleep. |

| | |
|---|---|
| **User** | What are your initial thoughts on this segment? |
| **CerebraGloss** | This segment appears to be largely unremarkable, showing a stable background rhythm. While there's some ~~high-frequency noise~~, particularly in the frontal and temporal regions, it doesn't obscure the overall tracing significantly. |
| **Answer Key** | This appears to be a good quality recording of a patient in sleep. The most prominent features are two well-formed, generalized bursts of sleep spindles, which are characteristic of NREM Stage 2 sleep. |

**Epileptiform discharge**

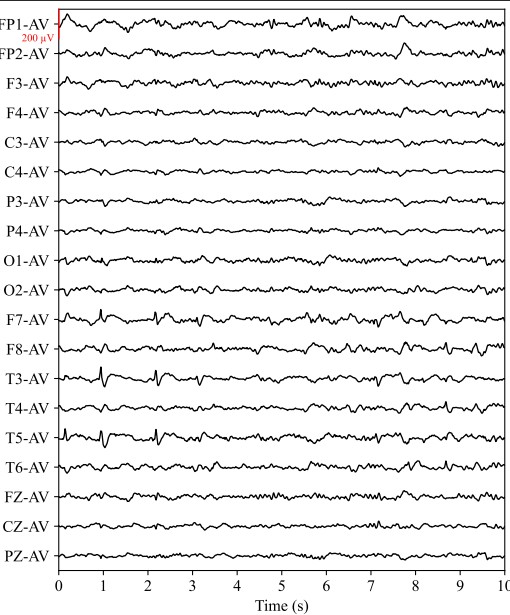

Source: `CerebraGloss-Bench/img/122_av.jpg`

| | |
|---|---|
| **User** | What is happening in this EEG? |
| **CerebraGloss** | The channel layout uses the standard 10-20 system, using average reference. Overall data quality is good. Sporadic sharps and spikes are present in bilateral frontal, temporal, and central lobes during 0.1-0.5s, 0.7-1.1s, 2.0-2.4s. Sporadic sharps are present in FP1, FP2, F8 during 7.3-7.9s. ~~Sporadic sharps are present in T5 during 8.6-9.0s. Delta activity is observed in F7, T3, T4 throughout the recording period, with frequency approximately 1.6 Hz and medium amplitude (37.2 µV).~~ The background activity is stable, predominantly composed of medium-amplitude Delta band (at approximately 2.1 Hz, amplitude about 25.8 µV). Interhemispheric asymmetry is observed in the background activity, manifested as: on F7/F8, T3/T4 leads, the amplitude on the left side is consistently higher than that on the right side. |
| **Answer Key** | The channel layout uses the standard 10-20 system with an average reference. The overall data quality is good. Sporadic independent spikes occur over the left anterior temporal region (F7, T3, T5) at 0, 1, and 2s; and the right temporal region (F8, T4, T6) at 8.7s. |
| **User** | What is the most accurate description of the epileptiform activity seen in this segment? 
 A) A single, well-defined left temporal seizure focus. 
 B) Generalized spike-and-wave discharges. 
 C) Focal, independent epileptiform discharges. 
 D) Benign Rolandic spikes. |
| **CerebraGloss** | **C.** The segment shows sporadic spikes and sharps, some localized to the left temporal region (T3, T5), and others more widespread, indicating focal, independent epileptiform activity rather than a single generalized or focal seizure. |
| **Answer Key** | **C.** The EEG shows distinct sharp waves and spikes occurring independently in both the left (F7/T3/T5) and right (F8/T4) temporal regions, which is the key finding. |
| **User** | Can you elaborate on the epileptiform discharges in this segment? |
| **CerebraGloss** | Certainly. There are periodic sharp waves and spikes observed bilaterally in the frontal, temporal, central, and parietal regions, occurring at approximately 0.6 Hz. Additionally, sporadic sharp waves are noted in the right frontal and temporal lobes, specifically around 7.4-7.9 seconds and 8.3-8.7 seconds. |
| **Answer Key** | Certainly. Early in the recording, there are clusters of spikes localized to the left temporal region at around 0s, 1s and 2s. Then, much later in the segment, a separate and distinct group of spikes appears over the right temporal area at around 8.7s. Their separation in time and distinct locations suggest two independent foci of irritability. |

---

**Delta activity**

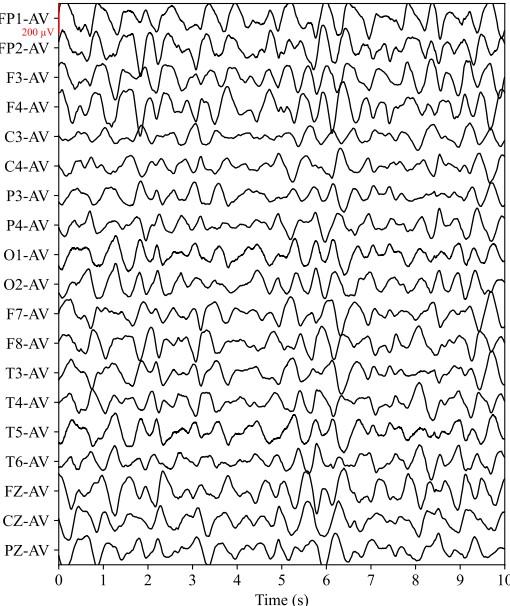

Source: `CerebraGloss-Bench/img/182_av.jpg`

| | |
|---|---|
| **User** | Provide a caption for this EEG segment. |
| **CerebraGloss** | The channel layout uses the standard 10-20 system with an average reference. The overall data quality is good. The recording is characterized by continuous, high-amplitude delta activity, approximately 1.3 Hz, observed diffusely across all brain regions. No definite epileptiform discharges or physiological sleep waves are observed. This pattern is consistent with a diffuse slowing of background activity. |
| **Answer Key** | The channel layout uses the standard 10-20 system with an average reference. The overall data quality is good. The background activity is stable and predominantly composed of high-amplitude, diffuse delta activity at approximately 1.8 Hz. If this is during sleep, it could be the EEG pattern of Stage N3 sleep. |

---

| | |
|---|---|
| **User** | The most prominent feature of this EEG segment is:
A) High-amplitude, generalized delta activity.
B) A focal epileptiform discharge in the right central region.
C) A normal, well-organized posterior dominant alpha rhythm.
D) Intermittent, rhythmic theta bursts. |
| **CerebraGloss** | **A.** The segment shows widespread, high-amplitude delta activity across all channels, indicating a generalized slowing of background activity. |
| **Answer Key** | **A.** |

---

| | |
|---|---|
| **User** | Can you give me an initial impression of this 10-second segment? |
| **CerebraGloss** | This segment appears to be within normal limits, showing a stable background dominated by delta activity. There are no clear epileptiform discharges or significant artifacts that would suggest pathology. |
| **Answer Key** | Certainly. The most striking feature of this segment is the background activity. It is quite slow, dominated by continuous, high-amplitude delta waves at approximately 1.8 Hz. The overall data quality appears good, which allows for a clear analysis. |

---

