# OpenReview forum: "CerebraGloss: Instruction-Tuning a Large Vision-Language Model for Fine-Grained Clinical EEG Interpretation"
_ICLR.cc/2026/Conference — ICLR 2026 Poster_

### Official Review · Reviewer_fjdA · 2025-10-29

**Soundness:** 3
**Presentation:** 3
**Contribution:** 2
**Rating:** 4
**Confidence:** 4

**Summary:**

CerebraGloss introduces an instruction-tuned large vision-language model (LVLM) for fine-grained clinical EEG interpretation. The paper contributes (i) a programmatic data engine that converts raw EEG into structured annotations via a bespoke YOLO-based waveform detector (CerebraGloss-YOLO) plus background/artifact analyzers; (ii) a two-stage post-training pipeline on top of Qwen2.5-VL (3B/7B) that first aligns EEG visual concepts and then performs instruction tuning for generative and conversational tasks; and (iii) CerebraGloss-Bench, a 90-segment, expert-verified benchmark assessing descriptions, complex MCQs, QA, and dense channel-wise waveform detection. Empirically, CerebraGloss surpasses general LVLMs (including GPT-5) on the new benchmark (e.g., MCQ 80.0%, ROUGE-1 44.19, QA 4.76/10 judged by GPT-5) and achieves new SOTA balanced accuracy on TUSZ seizure detection, while remaining competitive on HMC sleep staging. Ablations support the two-stage design (notably very short Stage-1 alignment) and suggest scaling benefits at 7B.

**Strengths:**

*  Reframes EEG analysis from narrow classification to unified generative interpretation with multi-turn dialogue; introduces a YOLO-based waveform detector tailored to channel-wise EEG graphoelements; offers a practical two-stage alignment/tuning recipe that is efficient (≈4h per stage) and scales to 7B.
*  Solid multi-facet evaluation: (i) CerebraGloss-Bench with free-text, MCQ, QA, and detection; (ii) standard clinical tasks (TUSZ/HMC); (iii) ablations on Stage-1 duration, Stage-2 composition, and model size. Clear reporting of balanced accuracy for imbalanced tasks and $mAP@0.5$ for detection.
*  Pipeline is clearly described with concrete examples (e.g., montage, artifacts, spindles, K-complexes), plus a concise clinical EEG primer that defines background rhythms and graphoelements. Limitations and ethics are explicitly discussed.
* The data engine and benchmark could catalyze research on general-purpose neuro-intelligent system*; reported SOTA on TUSZ indicates that the generative framing and instruction tuning can still deliver strong discriminative performance. Planned open-sourcing increases community value.

**Weaknesses:**

While the system is competently engineered, it largely follows a conventional recipe—fine-tuning a general-purpose foundation model on a domain-specific corpus with task-aligned instructions—without introducing a clearly new algorithmic idea. The components (instruction data curation, adapter/LoRA-style tuning, and standard loss/training schedules) appear incremental, and the paper does not articulate a principle or mechanism that would generalize beyond this application. As a result, the contribution feels more like a careful system instantiation than an advance in learning algorithms.

**Questions:**

1. Can the authors report per-class precision/recall (or error taxonomies) for **CerebraGloss-YOLO** against expert labels on a held-out set, and estimate the noise rate in the generated instruction data? This would contextualize hallucination rates.
2. Beyond GPT-5 judging, did clinicians perform blinded scoring (with inter-rater κ) on a subset of QA/description outputs? If not, could the authors add a small human study in the rebuttal?
5.  Can the model surface **calibrated uncertainty**, highlight **low-confidence regions**, or defer to human review automatically? This is especially important given acknowledged hallucinations and ethics.

---

> ### Author Response · Authors · 2025-11-19
>
> We sincerely thank the reviewer for their meticulous summary of our work and their insightful, high-level critique. We appreciate the acknowledgment of our solid evaluation, the reframing of the EEG analysis task, and the potential of our contributions to catalyze research. We understand the core concern regarding the nature of our contribution and offer the following points for consideration.
>
> **On the Central Weakness: Contribution as a "Conventional Recipe"**
>
> While we agree our work builds upon the powerful foundation model paradigm, we respectfully argue that framing it as an incremental application overlooks the core scientific and engineering challenges we had to overcome. The primary contribution of our work lies not in proposing a new standalone algorithm, but in providing a successful blueprint for adapting LVLMs to a complex, data-starved scientific domain where no such bridge previously existed. Of course, we have identified several shortcomings, including support for long time series, alignment between EEG signals (rather than images) and text, and the need for more robust data pipelines. Both in terms of architectural design and training strategies, these aspects provide ample room for improvement in future research.
>
> **On Question 1: Quantitative Analysis of Pipeline Noise**
>
> This is an excellent question. We have performed a detailed analysis of our pipeline's noise and its downstream impact.
>
> First, the per-class performance of our CerebraGloss-YOLO detector on the expert-labeled CerebraGloss-Bench is as follows (Average Precision):
>
> | Waveform   | AP   | Waveform  | AP   |
> | :--------- | :--- | :-------- | :--- |
> | `sharp`    | 0.63 | `eyem`    | 0.76 |
> | `spindle`  | 0.71 | `hfnoise` | 0.75 |
> | `Kcomplex` | 0.47 | `eyer+`   | 0.04 |
> | `spike`    | 0.19 | `eyer-`   | 0.15 |
> | `spsw`     | 0.0  |           |      |
>
> Second, we found a direct and quantifiable correlation between this detector noise and our final model's performance. An error analysis of CerebraGloss's MCQ performance reveals:
>
> *   **Sleep:** 4% error rate (relies on well-detected `spindle` waves, AP 0.71)
> *   **Epilepsy:** 33% error rate (relies on poorly-detected `spsw` and `spike` waves, AP 0.0 and 0.19)
>
> This strong correlation provides a clear, quantitative link between the noise rate in our generated data and the model's subsequent "hallucinations," directly contextualizing its failure modes. *We have added Appendix I to further discuss the issue.*
>
> **On Question 2 & 3: Clinical Validation and Uncertainty Quantification**
>
> These are two outstanding and critical questions for the clinical translation of any AI model.
>
> We wholeheartedly agree that a blinded scoring study by clinicians with inter-rater agreement and the implementation of calibrated uncertainty are vital steps for ensuring safety and reliability. However, our current work is positioned as a foundational study to establish technical feasibility. The primary goal was to prove that a generative model could learn the complex language of EEG interpretation at all.
>
> Conducting a formal clinical study and developing a robust uncertainty quantification framework are both significant research efforts in their own right. Given the promising results of this initial work, these are now our highest priorities for future research, and *we have explicitly added them to the future work section in our revised paper*.
>
> We hope these clarifications help to re-contextualize our work's contribution not as a simple application of a known recipe, but as a foundational effort that provides the data, methods, and insights necessary to open up a challenging clinical domain to the power of generative AI. We thank the reviewer again for their rigorous and thought-provoking feedback.

---

### Official Review · Reviewer_id5r · 2025-10-31

**Soundness:** 3
**Presentation:** 3
**Contribution:** 3
**Rating:** 6
**Confidence:** 4

**Summary:**

This paper introduces CerebraGloss, a large vision-language model (LVLM) instruction-tuned for clinical EEG interpretation. The key innovation is treating EEG waveforms as visual data and adapting LVLMs to interpret them through natural language. The authors develop an automated data generation pipeline featuring CerebraGloss-YOLO for waveform detection, create a large-scale instruction dataset (94K examples), and establish CerebraGloss-Bench as the first benchmark for open-ended EEG interpretation. The model achieves state-of-the-art performance on seizure detection (TUSZ) but not sleep staging (HMC) and shows convincing results on their novel benchmark (but no comparisons are available).

**Strengths:**

- Shifts from narrow classification to comprehensive, interpretative analysis of EEG, mimicking expert clinical practice
- Novel approach in treating EEG data as images
- CerebraGloss-Bench fills a critical gap with 90 expert-validated segments covering diverse clinical phenomena
- New SOTA results on seizure detection (but not sleep staging)
- Includes ablation studies on training stages, data composition, and model scaling
- Open-source commitment: Plans to release model, benchmark, and tools to advance the field
- Nice additional supplementary material for clinical context, more clinical examples and useful information on used prompts

**Weaknesses:**

- The practical interpretability of “neuro-glosses” remains somewhat subjective, and quantitative evaluation of explanation quality is limited.
- Primarily applies existing LVLM techniques (Qwen2.5-VL) to a new domain without substantial architectural innovations
- Treats EEG as images rather than native time-series signals (a comparison with methods operating directly on the time-series data would be helpful)
-  CerebraGloss-Bench might be too small with 90 expert-validated segments (for robust validation)
- Heavy reliance on automated pipeline introduces noise; authors acknowledge hallucination issues from false positives (also pointed out as limitation by the authors)
- Limited set of comparisons (regarding multimodal or EEG models)

**Questions:**

- Validation: Have you conducted any pilot studies with neurologists comparing CerebraGloss interpretations to expert readings? What is the inter-rater agreement between your automated annotations and expert clinicians on a held-out validation set?
- Have you conducted systematic error analysis on CerebraGloss-YOLO's outputs? What are the most common failure modes?
- Could you provide confusion matrices showing which waveform types are most frequently confused?
- Regarding potential hallucinations: Would incorporating uncertainty quantification improve clinical safety?
- Have you tested extending context beyond 10 seconds using sliding windows or hierarchical processing?
- Regarding ablation: Could you test the model without the Gemini-generated data to isolate the contribution of automated vs. LLM-augmented data?
- Why only 90 segments in CerebraGloss-Bench? Is this sufficient for robust evaluation?
- Further comparisons: Could you compare against multimodal medical models like Med-PaLM or BiomedGPT? What about domain-specific EEG models that don't use vision-language approaches?
- Can you elaborate a bit more on what specific challenges prevent direct time-series encoding and why you are in the favor of treating EEG data as images?

---

> ### Author Response · Authors · 2025-11-19
>
> We thank the reviewer for their positive and constructive feedback and their recognition of our work's novelty and commitment to open science. Below, we address the specific points raised.
>
> **On Weakness 1 & Question 1: Subjectivity and Clinical Validation**
>
> To clarify our current evaluation, model's performance is quantitatively measured against the the expert-authored ground truth. The reviewer rightly suggests pilot studies with neurologists to directly compare interpretations—the ultimate test for clinical utility. While our current work establishes foundational technical feasibility, such a study is a critical next step and *has been added to the future work section.*
>
> **On Weakness 2: Architectural Innovation**
>
> We agree that our work builds on an existing LVLM architecture. Our core contribution is introducing a new paradigm for EEG analysis by addressing key bottlenecks: the lack of fine-grained data and a suitable generative benchmark. This foundation enables future architectural innovations in the field.
>
> **On Weakness 5, Questions 2 & 3: Pipeline Noise Analysis**
>
> We performed a systematic error analysis of CerebraGloss-YOLO on our benchmark. Key failure modes involve rare and morphologically subtle waveforms:
>
> | Waveform   | AP   | Waveform  | AP   |
> | :--------- | :--- | :-------- | :--- |
> | `sharp`    | 0.63 | `eyem`    | 0.76 |
> | `spindle`  | 0.71 | `hfnoise` | 0.75 |
> | `Kcomplex` | 0.47 | `eyer+`   | 0.04 |
> | `spike`    | 0.19 | `eyer-`   | 0.15 |
> | `spsw`     | 0.0  |           |      |
>
> Low AP for `spsw` and `eyer` events stems from their rarity in training (4% and 0.6% of boxes, respectively) and subtle appearance. Confusion matrix cannot be used for object detection tasks. We are actively developing a larger, cleaner dataset to address these issues. *We add Appendix I to further discuss the issue.*
>
> **On Weakness 3 & Question 9: Image vs. Time-Series Modality**
>
> Our initial explorations did involve integrating time-series encoders directly with an LLM. However, we found the performance was poor. A possible requirement is an encoder that can align signals with fine-grained text—a major research challenge. Using image aligns the model's input with the exact modality used by human experts. It also allows us to leverage the phenomenal power of pre-trained LVLMs. Table 4 already includes comparisons with non-vision models (EEGNet, ELM-MIL, etc.) on classification tasks.
>
> **On Weakness 4 & Question 7: Benchmark Size**
>
> We agree 90 segments is modest. Our priority for CerebraGloss-Bench was quality and diversity over quantity. Each segment was expert-selected and annotated to cover a broad range of clinical scenarios (Fig 3). Given the labor-intensive process, we view this as a foundational tool for the community and hope it inspires future expansion.
>
> **On Question 4: Uncertainty Quantification for Hallucinations**
>
> We fully agree that Uncertainty Quantification (UQ) is critical for clinical safety. We view this as a two-step process: first, improving data quality to reduce hallucinations from pipeline noise; then implementing UQ methods. *This has been added to our future work discussion.*
>
> **On Question 5: Extending Context Beyond 10 seconds**
>
> The reviewer correctly notes the importance of long-range dependencies in clinical EEG. Our 10-second window was a deliberate simplification to first address fine-grained waveform grounding. This work sets the stage for future research into long-context EEG models.
>
> **On Question 6: Ablation on Gemini-Generated Data**
>
> | LLM-augmented data | TUSZ      | HMC       | MCQ       | Description | QA       |
> | :----------------- | :-------- | :-------- | :-------- | :---------- | :------- |
> | without            | 78.39     | 61.29     | 47.78     | 9.02        | 2.34     |
> | **with**           | **79.21** | **62.02** | **80.00** | **44.19**   | **4.76** |
>
> Without LLM-augmented data, the model loses open-ended generative ability, defaulting to MCQ format or generating gibberish (e.g., `**E**.**G**.**E**.**T*.**R**...`), as the remaining Stage 2 data is predominantly MCQ-based. *We have added this experiment to our ablation section.*
>
> **On Weakness 6 & Question 8: Further Comparisons**
>
> *We added comparisons with BioMedGPT and LLaVA-Med (Med-PaLM is unavailable) to our revised paper.* Results confirm these models cannot interpret EEGs without specific training:
>
> | Model               | MCQ       | Description | QA       | TUSZ      | HMC       |
> | :------------------ | :-------- | :---------- | :------- | :-------- | :-------- |
> | LLaVA-Med           | /         | 8.87        | 2.83     | 50.00     | 25.00     |
> | BioMedGPT           | /         | 11.82       | 1.29     | 50.00     | 25.00     |
>
> For example, when describing an EEG image from Table 1, BioMedGPT produced an irrelevant response: *"5 cases of intracranial hemorrhage: a grid-like inclusions in the form of a logarithmic scale plot..."

---

### Official Review · Reviewer_NtLM · 2025-10-31

**Soundness:** 2
**Presentation:** 3
**Contribution:** 3
**Rating:** 6
**Confidence:** 4

**Summary:**

The authors introduce CerebraGloss, a Large Vision-Language Model (LVLM) that is instruction-tuned for the interpretation of clinical EEG. The authors' core claim is that this model moves beyond the field's current focus on narrow classification tasks (like seizure detection) to enable unified, generative analysis. This allows the model to generate detailed waveform descriptions and engage in multi-turn, context-aware dialogue about an EEG segment.

To overcome the lack of suitable training data, the authors developed a novel, automated data generation pipeline to create a large-scale corpus of EEG-text instruction data. To evaluate the model's new capabilities, they also introduce and release CerebraGloss-Bench, a benchmark for open-ended EEG interpretation. The authors demonstrate that CerebraGloss outperforms 'off-the-shelf' LVLMs (including proprietary models) on this new benchmark and also performs strong classification on the TUSZ seizure detection classification task.

**Strengths:**

- Novel and valuable contribution: The paper aims to start a broader shift around machine learning for EEG. Moving from isolated classification to a unified, generative, and conversational model for EEG interpretation is a novel and interesting direction for the field - certainly worthy of exploration.
- Significant data generation effort: The development of the automated data generation pipeline, especially the bespoke detector for waveform events, represents a substantial engineering effort. This pipeline is a key contribution to addressing the primary bottleneck (lack of fine-grained, large-scale paired data) that has held back this line of research.
- New generative bnenchmark: The release of their benchmark is a welcome contribution. The field so far I believe entirely lacks benchmarks for evaluating generative and conversational abilities, and this dataset (though small) provides a much-needed tool for measuring progress in this new research direction.

**Weaknesses:**

- Limited classification evaluation: The paper introduces CerebraGloss as a model capable of both classification and generation. However, the classification evaluation seems limited. Given that the data generation pipeline sourced data from numerous public datasets (TUAB, TUEV, TUAR, etc.), it is surprising that the model's classification performance was not also benchmarked on tasks from these datasets (e.g., abnormality detection on TUAB or a more complex event classification from TUEV). This would have provided a much more comprehensive picture of its capabilities as a "unified" model.
- Missing comparisons: The related work section discusses prior work like Gijsen & Ritter (2025) but dismisses it, stating it "struggles to ground textual descriptions." It is unclear if this claim is based on a direct comparison or is a hypothesis. This work (together with EEG-CLIP (https://doi.org/10.3389/frobt.2025.1625731) although that one is 'concurrent' so comparisons cannot be requested, yet citation seems appropriate), seems like an important baseline. It is unclear why other existing models, which also bridge EEG and language, were not further discussed or included in the comparisons.
- Small scale of the new benchmark: While CerebraGloss-Bench is a really valuable type of contribution, its scale of only 90 segments seems small. This raises concerns about the statistical robustness of the evaluation results presented in Table 3 and how well they might generalize.

I would be happy to increase my score if the evaluations are extended to better support the claims about the classification ability of the model or if the paper's presentation is appropriately adapted to focus on the other strengths of the authors' work.

**Questions:**

- Why was the classification evaluation limited to TUSZ and HMC? Could the model's performance be reported on other standard tasks from the datasets you already used for data generation?
- Why do the authors not provide any comparisons with Gijsen & Ritter, 2025 (given that EEG-CLIP is technically 'concurrent work')? It seems the most relevant work to yours.
- Could you elaborate on the decision to limit CerebraGloss-Bench to 90 samples?  Given the small size, how confident are you in the benchmark's ability to robustly differentiate model performance?
- Do the authors have any future intentions to release any of the data annotations or to expand the benchmark?
- I'm wondering about the trade-off of presenting the EEG as an image versus operating in 'signal/timeseries' space. I presume it is necessary to move to images to be able to leverage the pretrained VLMs, but have the authors considered integrating a timeseries encoder with an LLM? I would not expect any analyses on this but it seems an important design decision about which it would be interesting to know the authors' reasoning.
- In the Stage 1 ablation study (Section 6.3), you find the optimal checkpoint is at just 0.05 epochs to avoid overfitting and "overwriting its powerful, pre-existing reasoning capabilities." This suggests the model starts to overfit almost immediately. Could you provide more intuition on why this overfitting on the "template-heavy captions" happens so rapidly, before even a single pass over the data? Is this visible in the model outputs?

---

> ### Author Response · Authors · 2025-11-19
>
> We sincerely thank the reviewer for their thoughtful and detailed feedback. We are encouraged that they found our work's direction "novel and interesting" and recognized our contributions in data generation and benchmarking. We address the specific points below and believe our clarifications and new results will strengthen the paper.
>
> **On Weakness 1 & Question 1: Limited Classification Evaluation**
>
> We thank the reviewer for this important question. Our decision to focus on TUSZ and HMC, rather than tasks like TUAB and TUEV, was a deliberate one.
>
> *   **TUAB:** The issue is **label-granularity mismatch**. A file-level "abnormal" label does not reliably apply to every 10-second segment within a long recording, making segment-level training on these labels problematic (see Appendix C, Fig 8).
> *   **TUEV:** The problem is **single-label simplification for co-occurring events**. Segments often contain multiple simultaneous events, but are assigned only one simplified label, failing to capture the signal's complexity (see Appendix C, Fig 9).
>
> Due to these issues, we selected benchmarks with reliable segment-level labels to ensure a fair evaluation of our model's capabilities.
>
> **On Weakness 2 & Question 2: Missing Comparisons**
>
> 1.  **Comparison with Gijsen & Ritter (2025):** We apologize for our potentially misleading description and have revised it. The core distinction is in research goals: their work targets representations for classification, while ours targets fine-grained grounding for generation. To provide a direct comparison, we evaluated their official ELM-MIL model on TUSZ. By training a classifier on their extracted features, their method achieves 78.27% balanced accuracy, on par with the SOTA. *We have added this to Table 4, noting that a full comparison is challenging due to differing modalities and preprocessing method.*
> 2.  **Citation of EEG-CLIP:** *We regret the oversight and have now cited and discussed EEG-CLIP, which was published very close to our submission date, in our related work section.*
>
> **On Weakness 3 & Question 3: Small Benchmark Scale**
>
> We agree that 90 segments is a modest number. However, our primary goal in creating CerebraGloss-Bench was to prioritize quality, diversity, and clinical relevance over sheer volume. Each segment was carefully selected and annotated by experts to cover a comprehensive range of EEG phenomena and reasoning tasks (as shown in Figure 3). Creating such a high-quality benchmark is an extremely labor-intensive process. We view it as a crucial, foundational tool for the community to begin rigorously evaluating generative models, and we hope it will inspire collaborative efforts to expand it in the future.
>
> **On Question 4: Future Plans for Data and Benchmark Expansion**
>
> Yes. We are actively collaborating with a hospital to create a new dataset of over 2,000 EEG samples with detailed waveform annotations, planned for release next year. While we cannot release our current training annotations (derived from public data), our open-source data engine already enables the community. We also plan to expand CerebraGloss-Bench in the future.
>
> **On Question 5: Rationale for Image vs. Time-Series Modality**
>
> This is an excellent and important question. Our initial explorations did involve integrating time-series encoders directly with an LLM. However, we found the performance was poor. A truly effective signal-language model would likely require a CLIP-style pre-training objective focused on grounding fine-grained textual concepts (e.g., "a sharp wave in channel FP1 at 3.2 seconds") to specific signal segments—a significant research challenge in itself.
>
> Our decision to use the image modality was a pragmatic one that offers both a high floor and a high ceiling. It aligns the model's input with the exact modality used by human experts, ensuring no information is lost from a clinical perspective. It also allows us to leverage the phenomenal power of pre-trained LVLMs.
>
> **On Question 6: Rapid Overfitting in Stage 1**
>
> We thank the reviewer for this sharp observation; the model is indeed highly sensitive here.
>
> *   0.05 epochs is not "instantaneous"—it represents 50 minutes of training on 8 A800 GPUs. This is a rapid but significant process.
> *   The learning of template-based descriptions and EEG terminology is achieved within just 0.05 epochs. Subsequent training does not intuitively result in the model generating template-like language more easily—since that has been already learned by 0.05 epochs—but rather manifested as increased hallucination. For example, the model misidentifies alpha rhythm as sleep spindles when given the input from Table 1:
>     > **User:** What are your thoughts on this segment?
>     >
>     > **Cerebragloss (0.2 epochs):** This segment appears to be of good quality and was recorded during sleep. I observe prominent sleep spindles, particularly in the frontal and temporal regions...

---

> > ### Comment · Reviewer_NtLM · 2025-11-25
> >
> > Thank you to the authors for their reply, the additional evaluations, and additions.
> >
> > I believe the paper provides a nice contribution, but still its main weakness remains the limited amount of evaluation. For TUEV, are not also channel-specific labels provided? Why does this not allow you to do multi-class prediction given a segment? (i.e. simultaneously predict all present events in the presented crop)

---

> > > ### Author Response · Authors · 2025-11-26
> > >
> > > This is a profound and sharp question. To explain why we did not perform this evaluation, we must first objectively examine the TUEV dataset structure and the problematic ways the community currently handles it.
> > >
> > > **1. The Reality of the TUEV Dataset**
> > > TUEV provides channel-specific annotations in `.rec` files (format: `channel, start_time, end_time, class`), covering six classes (SPSW, GPED, PLED, EYEM, ARTF, BCKG). Each label duration is strictly 1 second.
> > > Looking at the dataset itself, two intrinsic issues arise:
> > >
> > > *   **Sparsity:** The labels are extremely sparse, with a lot of actual events unannotated (a common issue in manual annotation).
> > > *   **Hierarchy Confusion:** The classes are not mutually exclusive. `SPSW` (spike/sharp) conceptually includes `GPED` and `PLED` (periodic discharges). Similarly, `ARTF` (artifact) conceptually includes `EYEM` (eye movement). Mixing these levels creates inherent ambiguity.
> > >
> > > **2. Flaws in Current Community Practice**
> > > The standard processing method in literature is coarse: researchers take the 1-second labeled event, expand it by 2 seconds on each side to create a 5-second clip, and assign the original label to the whole clip. This approach introduces four critical flaws:
> > >
> > > 1.  **Duplicate Samples:** If Channel 21 and Channel 14 both have Event 5 at the same timestamp, the dataset contains two identical input with the same label.
> > > 2.  **Conflicting Labels:** If Channel 3 has Event 1 and Channel 4 has Event 5 at the same timestamp, the dataset contains the exact same input labeled as two different classes.
> > > 3.  **Label Simplification:** Due to sparsity and the expansion window, a segment often contains multiple events (e.g., co-occurring artifacts and spikes), but is forced into a single-label classification task.
> > > 4.  **Context Dependency:** `GPED` and `PLED` are periodic. Defining them based on a short clip without broader context is clinically unsound.
> > >
> > > **3. Feasibility of the Proposed Evaluation**
> > > We considered the reviewer's suggestion: why not feed a 10s segment to CerebraGloss and ask, "What event is in Channel X at Time T?" This would theoretically bypass the sparsity issue. However, this is practically infeasible for four reasons:
> > >
> > > *   **No Fair Baseline:** To our knowledge, no existing model supports this specific "channel-time query" on TUEV.
> > > *   **Technical Incompatibility (Montage):** TUEV uses the ACNS TCP (bipolar) montage for its labels, whereas CerebraGloss is trained on an average reference. The visual patterns are fundamentally different.
> > > *   **No Specific Optimization:** During training, we did not teach the model concepts like `GPED` or `PLED`. Furthermore, while our instruction-following dataset contains some localization tasks, the model was not specifically trained or optimized to handle the precise "channel-time" querying required for this rigorous evaluation.
> > > *   **Dataset Ambiguity:** Even if we overcame the above, the boundary between `SPSW`，`GPED` and `PLED` in the ground truth is often inconsistent, making fine-grained evaluation unreliable.
> > >
> > > **Conclusion**
> > > The reviewer’s suggestion regarding localization (specifically querying what event occurred in a specific channel at a specific time) is an excellent idea. It offers a new perspective for future benchmarking. We sincerely thank you for this question, as it inspired us to think more deeply about the latent problems in TUEV and current community practices.

---

### Official Review · Reviewer_x7qx · 2025-11-01

**Soundness:** 3
**Presentation:** 3
**Contribution:** 3
**Rating:** 6
**Confidence:** 3

**Summary:**

This paper introduces CerebraGloss, an instruction-tuned Large Vision-Language Model (LVLM) for nuanced interpretation of clinical EEG waveforms. The authors propose a full data-engine pipeline that generates synthetic EEG-text instruction data. The paper also introduces CerebraGloss-Bench, a benchmark of 90 EEG segments annotated for four evaluation formats (description, QA, MCQ, waveform detection). Experiments show improvements over general LVLMs (e.g., GPT-5, Qwen2.5-VL-3B) and new state-of-the-art results on TUSZ seizure detection.

**Strengths:**

1. EEG interpretation is a clinically valuable yet underexplored problem in multimodal AI. The authors frame the shift from narrow classification to generative interpretation.

2. The automated data engine is an engineering contribution. The combination of YOLO-based detection, spectral feature extraction, and rule-based/LLM caption synthesis is practical for a data-scarce field.

**Weaknesses:**

1. Although the pipeline is innovative, the core dataset is automatically generated with only light manual supervision. This raises concerns about noise propagation and factual validity in fine-grained annotations.

2. The authors suggest “occasional hallucinations” in generated interpretations in ethics statement, but more quantitative analysis of annotation accuracy or comparison to human expert labels would increase confidence.

3. CerebraGloss-Bench contains only 90 test segments.

4. Comparisons are restricted to general-purpose LVLMs or LLaVA-Med which is relatively out-dated. Some more recent and stronger specialized LVLMs should be included for comparison.

**Questions:**

1. More analysis on the generated data can be conducted. For example, the quantitative analysis of annotation accuracy or comparison to human expert labels.

2. Some stronger specialized baselines can be included.

---

> ### Author Response · Authors · 2025-11-19
>
> We sincerely thank the reviewer for their valuable feedback and positive assessment of our work's motivation and contributions. The comments are very helpful, and we address them below.
>
> **On Weakness 1, 2 & Question 1: Data Quality, Noise, and Hallucination Analysis**
>
> We agree that understanding the quality and potential noise of our automatically generated data is crucial. We offer several points of clarification and analysis.
>
> 1.  **Clarification on "Comparison to Human Expert Labels":** We would like to respectfully clarify a potential misunderstanding. The results presented for CerebraGloss-Bench in our paper are a direct comparison against human expert annotations. CerebraGloss-Bench itself, including all descriptions, QA pairs, MCQs, and bounding boxes, was meticulously created by human experts. Therefore, the performance reported in Tables 2 and 3 already represents a quantitative comparison of our model's outputs against a human-defined gold standard.  *We have revised the relevant paragraph to make it clearer. Additionally, we fully endorse the approach of incorporating more human expert evaluation and have included it in the revised "Future Work" section.*
>
> 2.  **Quantitative Analysis of Pipeline Noise:** To quantify the noise from our data pipeline, we evaluated our YOLO-based waveform detector on the human-annotated CerebraGloss-Bench. The Average Precision (AP) scores are:
>
> | Waveform   | AP   | Waveform  | AP   |
> | :--------- | :--- | :-------- | :--- |
> | `sharp`    | 0.63 | `eyem`    | 0.76 |
> | `spindle`  | 0.71 | `hfnoise` | 0.75 |
> | `Kcomplex` | 0.47 | `eyer+`   | 0.04 |
> | `spike`    | 0.19 | `eyer-`   | 0.15 |
> | `spsw`     | 0.0  |           |      |
>
> The detector performs well on common waveforms but struggles with rare or subtle ones. Crucially, this reveals a direct correlation: our model's downstream performance mirrors the detector's weaknesses. For example, CerebraGloss answered 6 of 18 multiple-choice questions on epilepsy incorrectly, corresponding to the detector's low AP on `spike` and `spsw`. Conversely, it correctly answered 22 of 23 multiple-choice questions on sleep, corresponding to high AP on `spindle`. *We have added Appendix I to further discuss the issue.*
>
> To address this foundational challenge, we are actively collaborating with local hospitals to develop and release a large-scale, expert-annotated waveform dataset next year. This will allow for the training of more robust detectors and further enhance the quality of the entire pipeline.
>
> **On Weakness 3: Benchmark Size**
>
> We acknowledge that CerebraGloss-Bench's size of 90 segments is modest. However, its creation was a labor-intensive process requiring deep clinical expertise for data selection and annotation. Our priority was comprehensiveness and diversity over sheer volume. As shown in Figure 3, these 90 segments were carefully curated to cover a wide and balanced distribution of clinical scenarios and reasoning tasks. We view it as a robust, foundational resource for the community to build upon.
>
> **On Weakness 4 & Question 2: Comparison with Stronger Specialized Baselines**
>
> We thank the reviewer for this excellent suggestion. In response, we have conducted new experiments with more recent and specialized biomedical LVLMs, BioMedGPT (Luo et al., 2024), and have also completed the evaluation for LLaVA-Med on CerebraGloss-Bench. The results are as follows, *which is also added to Table 3 and 4 in our revised paper*:
>
> | Model               | MCQ       | Description | QA       | TUSZ      | HMC       |
> | :------------------ | :-------- | :---------- | :------- | :-------- | :-------- |
> | LLaVA-Med           | /         | 8.87        | 2.83     | 50.00     | 25.00     |
> | BioMedGPT           | /         | 11.82       | 1.29     | 50.00     | 25.00     |
> | **CerebraGloss-3B** | **80.00** | **44.19**   | **4.76** | **79.21** | **62.02** |
>
> Neither BioMedGPT nor LLaVA-Med could follow the multiple-choice question (MCQ) format instructions, resulting in null scores. More importantly, both models failed to interpret EEG content meaningfully. For instance, when tasked with describing an EEG image from Table 1, BioMedGPT generated a completely irrelevant response: "5 cases of intracranial hemorrhage : a grid-like inclusions in the form of a logarithmic scale plot, which demonstrates the reliability of the scoring system."
>
> These results highlight a critical point: interpreting highly specialized modalities like EEG requires domain-specific data exposure. General biomedical LVLMs, despite their broader knowledge, fail to ground their outputs without it. This underscores the necessity and significance of our data generation pipeline and fine-tuning approach.
>
> We hope these responses have thoroughly addressed the reviewer's concerns. We thank you again for your constructive engagement.

---

### Official Review · Reviewer_8gRC · 2025-11-11

**Soundness:** 2
**Presentation:** 4
**Contribution:** 4
**Rating:** 6
**Confidence:** 3

**Summary:**

This paper presents CerebraGloss, the first Large Vision-Language Model (LVLM)  fine-tuned for clinical EEG interpretation through instruction tuning. It transforms the traditional EEG interpretation task from a closed classification problem into a generative, multi-turn conversational explanation task, supported by an automated EEG–text instruction data generation pipeline and a new evaluation benchmark called CerebraGloss-Bench. The model demonstrates strong performance on standard datasets (TUSZ, HMC) and a custom benchmark, showing promising potential for both research and practical applications.

**Strengths:**

CerebraGloss represents an important step toward generative and multi-task EEG interpretation empowered by LVLMs

**Weaknesses:**

Data quality and validation are insufficient: The impact of noise in the automated data pipeline is not quantitatively analyzed. The paper lacks comparison with models trained on manually annotated data, making it unclear how noise affects performance. The representativeness of the data distribution is also not well explained—although the training data come from several public datasets, it is uncertain whether they cover different ages and pathological EEG types. The model’s generalization ability on heterogeneous data remains questionable.


Methodological depth could be improved: Converting EEG signals into images may lose important phase and amplitude dynamics, reducing diagnostic fidelity. The instruction tuning strategy heavily depends on an existing framework (Qwen2.5-VL), with limited architectural innovation introduced by the paper itself.


Writing style and tone are somewhat promotional: Some phrases sound overly promotional, such as “we pioneer” and “significantly outperforms GPT-5”. There are also minor spelling errors (e.g., “Descripition” should be “Description”).

**Questions:**

Evaluation lacks objectivity and completeness: The dialogue evaluation relies on GPT-5 scoring, which may introduce subjectivity and bias. No human expert evaluation is provided, making it hard to assess the true clinical relevance of the generated content. In addition, CerebraGloss-Bench contains only 90 EEG segments, which limits the statistical significance of the results.

**Details Of Ethics Concerns:**

a new dataset for EEG

---

> ### Author Response · Authors · 2025-11-19
>
> We sincerely thank the reviewer for their diligent work and constructive feedback. The comments are insightful and have helped us significantly clarify our contributions. We address each concern below.
>
> **On Weakness 1: Data Quality and Validation**
>
> We thank the reviewer for this critical point. The scarcity of large-scale, manually annotated EEG-text data is the primary bottleneck our work aims to address. Our automated pipeline is a first step toward solving this.
>
> 1.  **Noise Analysis:** We agree a quantitative analysis is crucial. Unluckily, we have to admit that we cannot compare with models trained on manually annotated data because such data does not exist. However, we have conducted further analysis on noise and model hallucinations. The Average Precision (AP) of CerebraGloss-YOLO are:
>
> | Waveform   | AP   | Waveform  | AP   |
> | :--------- | :--- | :-------- | :--- |
> | `sharp`    | 0.63 | `eyem`    | 0.76 |
> | `spindle`  | 0.71 | `hfnoise` | 0.75 |
> | `Kcomplex` | 0.47 | `eyer+`   | 0.04 |
> | `spike`    | 0.19 | `eyer-`   | 0.15 |
> | `spsw`     | 0.0  |           |      |
>
> The detector performs well on common waveforms but struggles with rare or subtle ones. Crucially, this reveals a direct correlation: our model's downstream performance mirrors the detector's weaknesses. For example, CerebraGloss answered 6 of 18 multiple-choice questions on epilepsy incorrectly, corresponding to the detector's low AP on `spike` and `spsw`. Conversely, it correctly answered 22 of 23 multiple-choice questions on sleep, corresponding to high AP on `spindle`. *We have added Appendix I to further discuss the issue.*
>
> 2.  **Data Representativeness:** The distribution covers a wide range of age groups, with a strong representation from adult to elderly populations:
>     `[0-20): 2.6%`, `[20-40): 23.4%`, `[40-60): 40.8%`, `[60-80): 28.3%`, `[80-100): 4.9%`.
>     Regarding pathology, clinical EEG analysis focuses on identifying fundamental waveforms rather than specific disease labels. Our dataset is built on this principle, ensuring it covers a comprehensive set of elemental patterns essential for clinical diagnosis. To further improve data quality, we are collaborating with hospitals to build and release a larger, expert-annotated waveform dataset next year.
>
> **On Weakness 2: Methodological Depth**
>
> 1.  **EEG-to-Image Conversion:** Our conversion of EEG signals to images directly mimics standard clinical practice. Neurologists interpret these exact visual formats, which preserve all critical signal properties (e.g., phase, amplitude, morphology). This approach aligns our model's input modality with expert workflow, allowing us to leverage the powerful reasoning of modern LVLMs.
>
> 2.  **Reliance on Existing Frameworks:** Our primary goal was to establish a new paradigm for generative EEG interpretation, not to design a novel architecture. By building on a strong foundation model, we could focus on the domain's core bottlenecks: data scarcity and the lack of an appropriate benchmark. Our key contributions—the automated data pipeline and CerebraGloss-Bench—are foundational steps that enable future research into specialized architectures for neuro-intelligent systems.
>
> **On Weakness 3: Writing Style**
>
> We appreciate the feedback and have revised the manuscript accordingly. *We have adopted a more objective tone, removing promotional phrases like "we pioneer," and have corrected all identified spelling errors (e.g., "Descripition").*
>
> **On Question 1: Evaluation Objectivity**
>
> We apologize for the lack of clarity regarding our evaluation protocol. The reviewer's concern is valid. Our process is not a subjective, free-form evaluation. GPT-5 serves as a structured scorer, not an open-ended judge. For each question, it is provided with the model's output, a human expert-written answer key, and a strict scoring rubric. Its task is to compare the model's response against this gold standard. This LLM-as-a-Judge method, when anchored to expert knowledge, is an established and scalable evaluation paradigm (Zheng et al., NeurIPS 2023). *We have revised the relevant paragraph to make it clearer. Additionally, we fully endorse the approach of incorporating more human expert evaluation and have included it in the revised "Future Work" section.*
>
> **On Question 2: Benchmark Size**
>
> We acknowledge that CerebraGloss-Bench's size of 90 segments is modest. However, its creation was a labor-intensive process requiring deep clinical expertise for data selection and annotation. Our priority was comprehensiveness and diversity over sheer volume. As shown in Figure 3, these 90 segments were carefully curated to cover a wide and balanced distribution of clinical scenarios and reasoning tasks. We view it as a robust, foundational resource for the community to build upon.
>
> We hope these clarifications address the reviewer's concerns and further highlight the value of our work. We thank you again for your time and consideration.

---

### Author Response · Authors · 2025-11-28
**Global Response**

We thank all reviewers (8gRC, x7qx, NtLM, id5r, fjdA) for their time and insightful feedback. We are encouraged that the reviewers recognized the **novelty and value** of shifting EEG analysis from narrow classification to unified generative interpretation (8gRC, x7qx, NtLM, id5r, fjdA), the **significance of our automated data engine** in addressing data scarcity (x7qx, NtLM, id5r), and the **clinical relevance** of our approach (fjdA, id5r).

We have uploaded a revised manuscript. Below, we address the common concerns raised across reviews and summarize our updates.

**1. Quantitative Analysis of Pipeline Noise & Hallucinations (Response to 8gRC, x7qx, id5r, fjdA)**

Reviewers rightly pointed out the need to quantify the noise in our automated pipeline and its impact on model hallucinations.

*   **New Analysis:** We have added **Appendix I**, which details the Average Precision (AP) of `CerebraGloss-YOLO` and the error rates of CerebraGloss on CerebraGloss-Bench MCQs
*   **Key Insight:** We found a direct correlation between detector performance and downstream model hallucinations. For example, the model performs excellently on sleep staging (where `spindle` detection AP is 0.71) but struggles with specific epilepsy subtypes (where `spike` detection AP is 0.19). This quantitative link confirms that improving the detector (via our future expert-annotated dataset) is the clear path to reducing hallucinations.

**2. Benchmark Scale (Response to 8gRC, x7qx, NtLM, id5r)**

Reviewers noted that CerebraGloss-Bench (90 segments) is small.

*   **Clarification:** We prioritized **quality, diversity, and expert validation** over quantity. As shown in Figure 3, these segments were meticulously curated to cover a comprehensive distribution of clinical scenarios. Constructing such a benchmark is labor-intensive, involving expert selection and gold-standard annotation.
*   **Value:** We view this benchmark as a "seed" for the community—a high-quality evaluation standard that sets the precedent for generative EEG assessment, which we and the community can expand upon.

**3. EEG-to-Image Modality & Architectural Novelty (Response to 8gRC, NtLM, id5r, fjdA)**

Some reviewers questioned the use of images over time-series and the reliance on existing LVLM architectures.
*   **Modality:** Converting EEG to images aligns the model's input with the **exact modality used by clinical experts** (visual interpretation), ensuring no loss of diagnostic information (phase/amplitude are visible). Furthermore, currently, no time-series encoder exists that can effectively ground fine-grained textual concepts (e.g., "spike at 3s") as well as vision-language models can.
*   **Architecture:** Our contribution is not a new Transformer block, but a **new paradigm**. We provide the first successful blueprint (Data Engine + Instruction Tuning) to bridge the gap between EEG and LVLMs, which may offer insights for other data-scarce domains. We also find some intriguing results: for example, early stopping at a loss of 1.6 shows the best performance.

**4. Additional Baselines (Response to x7qx, id5r)**

To further validate our performance, we conducted new experiments with **BioMedGPT** and **LLaVA-Med** (added to Tables 3 & 4).

*   **Result:** These models failed to produce meaningful EEG interpretations or follow MCQ instructions (Score: ~0).
*   **Conclusion:** This underscores that general biomedical knowledge is insufficient for EEG interpretation. Our domain-specific data engine and instruction tuning are essential for success in this modality.

**Summary of Revisions in the Manuscript:**

1.  **Added Appendix I:** Detailed noise analysis of the data pipeline and error analysis of the model.
2.  **Updated Tables 3, 4 & 5:** Included results for BioMedGPT, LLaVA-Med, ELM-MIL and ablation for LLM-augmented data.
3.  **Revised Future Work:** Explicitly added plans for Uncertainty Quantification (UQ) and formal Clinical Validation studies.
4.  **Clarifications:** Corrected the description of related work (Gijsen & Ritter), clarified the GPT-5 scoring rules, cited EEG-CLIP, and improved the objectivity of the writing tone.

We hope these responses and revisions address your concerns. We believe CerebraGloss represents a foundational step toward general-purpose neuro-intelligent systems, **returning EEG analysis to its rich, clinical value**.

---

### Author Response · Authors · 2025-12-01
**Summary Comment (1/2)**

Dear Area Chair,

Given the reassignment due to the recent disruptions, we provide this summary to facilitate your review. It outlines our paper’s contributions, how we addressed reviewer concerns (including new experiments and analyses), and a summary of the active discussion with Reviewer NtLM.

### 1. Paper Summary and Contributions

**Goal:** Current EEG analysis models are limited to narrow classification (e.g., seizure detection). We aim to shift the paradigm to **holistic, generative interpretation** (e.g., detailed description of all events, multi-turn dialogue between users and the model), mimicking clinical experts. For example, the model should be able to explain to the user that it identified NREM Stage 2 sleep by detecting sleep spindles in the vertex region over 1–2 seconds, rather than merely providing the answer of NREM Stage 2.

**Method:** We introduce **CerebraGloss**, an instruction-tuned Large Vision-Language Model (LVLM) for fine-grained clinical EEG interpretation.

*   **Data Engine:** To address the scarcity of fine-grained EEG-text pairs, we developed an automated pipeline featuring a custom **YOLO-based waveform detector** to programmatically generate a large-scale instruction dataset from raw EEG signals.
*   **Model:** We fine-tuned a general-purpose LVLM (Qwen2.5-VL) to "read" EEG images and generate clinical dialogue.

**Contributions:**
1.  **New Paradigm:** Shifting EEG analysis from isolated classification to unified generative dialogue.
2.  **Automated Data Pipeline:** A novel method to generate large-scale, fine-grained EEG instruction data.
3.  **CerebraGloss-Bench:** The first expert-curated benchmark (90 segments) for open-ended EEG interpretation.
4.  **Performance:** Achieved SOTA on TUSZ (seizure detection) and outperformed proprietary models (e.g., GPT-5) on our generative benchmark. We commit to open-sourcing the model, benchmark, and data tools.

### 2. Reviewers' Comments and Our Revisions

We received constructure feedback from 5 reviewers (8gRC, x7qx, NtLM, id5r, fjdA). All recognized the **novelty** of the generative approach and the **value** of the data engine. We have addressed all the concerns raised by the reviewers, which are summarized below:

*   **Concern 1: Pipeline Noise & Hallucinations (8gRC, x7qx, id5r, fjdA)**
    *   *Critique:* Need to quantify noise in the automated data generation and its impact on model hallucinations.
    *   *Action:* We added **Appendix I** with a quantitative analysis. We found a direct correlation: the model performs well on waveforms where our YOLO detector has high Average Precision (e.g., Spindles, AP 0.71) but hallucinates on classes where detector AP is low (e.g., Spikes, AP 0.19). This confirms that improving the detector (our next step) will reduce hallucinations.
*   **Concern 2: Benchmark Size (8gRC, x7qx, NtLM, id5r)**
    *   *Critique:* 90 segments in CerebraGloss-Bench seem small.
    *   *Response:* We prioritized **quality and expert validation** over quantity. Each segment was manually curated to cover diverse clinical scenarios. It serves as a high-quality "seed" for the community to expand.
*   **Concern 3: EEG-to-Image Modality & Architectural Novelty (8gRC, NtLM, id5r, fjdA)**
    *   *Critique:* Reviewers questioned the decision to convert EEG to images (vs. time-series) and noted the reliance on existing LVLM architectures without a new model block design.
    *   *Response:* regarding **modality**, converting EEG to images aligns with the **clinical gold standard** (neurologists read images) and leverages the superior grounding capabilities of VLMs, as no current time-series encoder can effectively align signals with fine-grained text. Regarding **architecture**, we argue our contribution is not a new Transformer block, but a **new paradigm**. We provide the first successful blueprint (Data Engine + Instruction Tuning) to bridge the gap between EEG and LVLMs in a data-scarce domain. This approach establishes a baseline for the field and offers valuable insights into training dynamics (e.g., specific early stopping behaviors) for future specialized architectures.
*   **Concern 4: Comparisons & Baselines (x7qx, id5r, NtLM)**
    *   *Critique:* Request for more specialized baselines.
    *   *Action:* We added experiments with **BioMedGPT** and **LLaVA-Med** (Tables 3 & 4). Both failed to interpret EEG meaningfully, proving that general biomedical knowledge is insufficient and our domain-specific instruction tuning is essential. We also added comparisons to **ELM-MIL** as requested.

---

> ### Author Response · Authors · 2025-12-01
> **Summary Comment (2/2)**
>
> *   **Other Comments & Revisions:**
>     *   Added **Uncertainty Quantification** and **Clinical Validation** plans to the "Future Work" section. (8gRC, id5r, fjdA)
>     *   Clarified the GPT-5 scoring rules. (8gRC)
>     *   Refined the tone to be more objective and fixed spelling mistakes. (8gRC)
>     *   Explained the data representativeness about age and pathology. (8gRC)
>     *   Clarified the distinction from concurrent work. (NtLM)
>     *   Explained early stop in stage 1. (NtLM)
>     *   Added ablation study for LLM-augmented data. (id5r)
>     *   Explained the choice of 10-second input. (id5r)
>
> ### 3. Summary of Discussion with Reviewer NtLM
>
> Reviewer NtLM was the only reviewer who engaged in the discussion phase and expressed their happiness to raise the score.
>
> *   **Topic:** NtLM asked why we limited classification evaluation to TUSZ/HMC and did not include other datasets like TUAB or TUEV.
> *   **Our Response:** We explained that TUAB labels are file-level (too coarse for segment training) and TUEV labels suffer from single-label simplification for co-occurring events.
> *   **Follow-up:** NtLM suggested a specific evaluation format for TUEV. We regarded it as asking the model "What event is in Channel X at Time T?".
> *   **Final Discussion:** We provided a detailed analysis of why this is practically infeasible including intrinsic flaws (e.g., conflicting labels for identical inputs, ambiguity between hierarchical classes like Spike vs. GPED), the lack of baselines supporting channel-time queries and different montages. We agreed that this idea offers a great perspective for *future* benchmark construction, but the current dataset limitations prevent a fair comparison.
>
> ### 4. Conclusion
>
> CerebraGloss represents a foundational step in bringing the capabilities of LVLMs to clinical neurophysiology. We have addressed the reviewers' concerns by **quantifying pipeline noise**, **adding biomedical LVLM baselines**, and **clarifying our design choices** regarding modality and datasets.
>
> We believe our work provides the necessary tools (Data Engine) and evaluation standards (Benchmark) to catalyze research into general-purpose neuro-intelligent systems. We are ready to open-source our assets to the community. You can refer to the video demo and the corresponding code in our supplementary materials, which were submitted as early as the initial paper submission.
>
> Thank you for your time and supervision.
>
> Sincerely,
> The Authors

---

### Meta-Review · Area_Chair_4j33 · 2026-01-09

**Summary:**

The paper introduced CerebraGloss, an instruction-tuned VLM for interpretation of clinical EEG waveforms. The authors introduced an automatic data generation pipeline, YOLO-based waveform detector, and an EEG-text instruction tuning dataset to facultative the VLM training. CerebraGloss demonstrates strong performance, surpassing leading LVLMs, including proprietary models like GPT-5. The authors promised to open-source their model, benchmark, and tools which will be a valuable resource to support the future research on this topic. Before rebuttal, 3 reviewers suggested marginally above acceptance and one reviewer suggested marginally below acceptance. The authors have addressed most of the concerns raised by the reviewers.

**Reviewer Concerns:**

Addressed

- Data Representativeness
- EEG-to-Image Conversion
- Evaluation Objectivity and Benchmark Size
- Additional Baselines
- Quantitative Analysis of Pipeline Noise and Hallucinations

Outstanding
-Methodological Depth and Reliance on Existing Frameworks- I agree with the reviewer that proposing a new architecture or a clearer methodological contribution would have made the paper stronger. However, the importance and relevance of the proposed problem formulation and the annotation efforts override this limitation.

-Data Quality and Validation- This is only partially addressed in the rebuttal

**Reviewer Scores:**

As the authors have addressed most of the reviewers’ concerns during the rebuttal, I assume that all positive reviewers would have remained at their current marginally above-acceptance rating or slightly increased it to “accept (poster).” Since the key issue raised by reviewer fjdA is the lack of a core algorithmic contribution—which is still not resolved—I assume that reviewer fjdA may maintain their current marginally below-acceptance rating. However, the AC has personally read the paper and agrees with the positive reviewers that the proposed benchmark, manually labeled datasets, YOLO-based elementary waveform detection, and the proposed model are valuable additions to the community. Hence, the AC recommends accepting the paper.

---

### Decision · Program_Chairs · 2026-01-26

Accept (Poster)